# Band Selection via Band Density Prominence Clustering for Hyperspectral Image Classification

**Chein-I Chang** [1,2,3,*], **Yi-Mei Kuo** [2] and **Kenneth Yeonkong Ma** [2]

1 Center for Hyperspectral Imaging in Remote Sensing (CHIRS), Information and Technology College, Dalian Maritime University, Dalian 116026, China

2 Remote Sensing Signal and Image Processing Laboratory, Department of Computer Science and Electrical Engineering, University of Maryland, Baltimore County (UMBC), Baltimore, MD 21250, USA; ccgsh10923@gmail.com (Y.-M.K.); a0721988@gmail.com (K.Y.M.)

3 Department of Electrical Engineering, National Cheng Kung University, Tainan 70101, Taiwan

* Correspondence: cchang@umbc.edu

**Abstract:** Band clustering has been widely used for hyperspectral band selection (BS). However, selecting an appropriate band to represent a band cluster is a key issue. Density peak clustering (DPC) provides an effective means for this purpose, referred to as DPC-based BS (DPC-BS). It uses two indicators, cluster density and cluster distance, to rank all bands for BS. This paper reinterprets cluster density and cluster distance as band local density (BLD) and band distance (BD) and also introduces a new concept called band prominence value (BPV) as a third indicator. Combining BLD and BD with BPV derives new band prioritization criteria for BS, which can extend the currently used DPC-BS to a new DPC-BS method referred to as band density prominence clustering (BDPC). By taking advantage of the three key indicators of BDPC, i.e., cut-off band distance $b_c$, $k$ nearest neighboring-band local density, and BPV, two versions of BDPC can be derived called $b_c$-BDPC and $k$-BDPC, both of which are quite different from existing DPC-based BS methods in three aspects. One is that the parameter $b_c$ of $b_c$-BDPC and the parameter $k$ of $k$-BDPC can be automatically determined by the number of clusters and virtual dimensionality (VD), respectively. Another is that instead of using Euclidean distance, a spectral discrimination measure is used to calculate BD as well as inter-band correlation. The most important and significant aspect is a novel idea that combines BPV with BLD and BD to derive new band prioritization criteria for BS. Extensive experiments demonstrate that BDPC generally performs better than DPC-BS as well as many current state-of-the art BS methods.

**Keywords:** band density prominent peak clustering (BDPC); band distance (BD); band local density (BLD); band prominence value (BPV); band selection (BS); hyperspectral image classification (HSIC); $k$ nearest neighbors ($k$NNs); shared nearest neighbor (SNN)

## 1. Introduction

With advances in remote sensing technology, hyperspectral imaging has become an emerging technique in recent years to solve many issues that cannot be resolved by multi-spectral imaging, such as subpixel detection, mixed pixel classification, spectral unmixing, endmember finding, etc. [1]. However, such advantages also come with unnecessary abundant spectral redundancy that can be removed without compromising data exploitation. One effective means is data spectral dimensionality reduction (DR) [2] (chapter 6). Although many DR techniques have been proposed over the past years, band selection (BS) remains one of most widely used DR methods for hyperspectral data reduction due to the fact that it can remove inter-band correlation resulting from very fine spectral resolution while retaining data integrity provided by selected bands.

### 1.1. Band Priorirtization for BS

Many BS methods have been developed in the past. Generally speaking, BS can be performed by band prioritization (BP), which ranks all bands according to a BP criterion. As a result, BP can be considered as unsupervised BS without requiring prior knowledge, selecting bands in accordance with their priority scores with no reference to a specific application, such as variance, entropy, information divergence, maximum variance principal component analysis (MVPCA) [3], mutual information [4,5], constrained band selection (CBS) [1,6], minimum estimated abundance covariance (MEAC) [7], etc. This type of BP is easy to implement and has been extensively used for BS.

### 1.2. Band Selection via Search Strategies

As an alternative to BP, another type of BS involves selecting desired bands according to a band search strategy, usually determined by a particular application, such as detection, classification, spectral unmixing, etc. Consequently, such BS is generally supervised and carried out by a specifically designed band search algorithm, such as sequential forward selection (SFS) algorithm [8], sequential backward selection (SBS) algorithm, multitask sparsity pursuit (MTSP) [9], multigraph determinantal point process (MDPP) [10], dominant set extraction BS (DSEBS) in [11], band subset (BSS) [12–14] for anomaly detection, linearly constrained minimum variance-based BSS for classification [15], and also evolution-based algorithms [16], using a variety of evolutionary computing strategies; for example, particle swarm optimization (PSO) [17,18], firefly [19], and colony algorithm [20,21].

### 1.3. Band Clustering/Grouping for BS

In addition to the above-mentioned BP-based and BS strategy-based BS methods, band clustering/grouping-based BS methods have also received considerable interest in the past. For example, Wang et al. [22] developed an adaptive subspace partition strategy (ASPS)-based band selection method by partitioning a hyperspectral image cube into multiple sub-cubes by maximizing the ratio of interclass distance to intra-class distance. In a follow-up work, Wang et al. [23] further developed a fast neighborhood grouping method for hyperspectral band selection (FNGBS), which partitions a hyperspectral image cube into several groups using a coarse-to-fine strategy, so that bands with the maximum product of local density and information entropy in groups are selected as a band subset. Most recently, Wang et al. [24] also developed an optimal neighborhood reconstruction (ONR)-based BSS based on correlated neighborhood property (CNP) to exploit strong correlations between neighboring bands. Interestingly, Baisantry et al. [25] developed a feature extraction-based clustering BS method that uses sparse subspace clustering to cluster similar bands into groups and prioritizes bands by a new metric called the combined divergence-correlation index to select the most discriminative as well as least correlated bands as cluster representatives.

### 1.4. Density Peak Clustering for BS

As noted above, band clustering/grouping-based methods cluster or group bands according to band neighboring correlation without taking data spatial distribution into account. To mitigate this dilemma, a new concept of density peak clustering (DPC) was derived in [26], which introduces two indicators, local density that characterizes the spatial distribution surrounding each data point and cluster distance of a data point from clusters with densities higher than its density. By virtue of DPC, we not only can calculate the density of each data point but also its distance to higher density clusters. Now, if we consider data points as spectral bands, then for each spectral band, say $\mathbf{B}_i$ (with its expanded band vector, denoted by $\mathbf{b}_i$), the data density can be interpreted as band local density (BLD), denoted by $\rho_i$, which is determined by the cut-off band distance (BD), denoted by $b_c$, and the cluster distance as BD, denoted by $\delta_i$ of $\mathbf{b}_i$, from band images with their BLD higher than $\rho_i$. Then, a DPC score is calculated by multiplying $\rho_i$ with $\delta_i$, referred to as $\gamma_i = \rho_i \times \delta_i$, to be used as a BP criterion to rank all bands for BS, referred to as DPC-BS.

*1.5. Motivations*

Although the above-mentioned DPC-BS methods have shown some success in BS, three issues still remain. One is how to automatically determine the cut-off band distance $b_c$, which is crucial to calculate the BLD. It must be determined in advance or empirically. Another is the used Euclidean distance, which generally measures the spherical distribution of the data so that a data point is always assigned to the nearest center. However, as for the bands of hyperspectral data, a better BD indicator should be a spectral discrimination measure that captures spectral characteristics better than distance. Last but not least, there is a lack of criteria for measuring inter-band correlation.

The motivation of this paper is to investigate an approach that can resolve the above three issues. More specifically, we develop a novel DPC-BS method called band density prominence clustering (BDPC), which introduces a new concept of band prominence value (BPV) for each band, say $\text{BPV}(\mathbf{b}_i)$, as a third indicator to take care of inter-band correlation. Despite the fact that this idea is inspired from the band prominent peaks of a curve calculated by self-mutual information (SMI) [5], it does not calculate prominent peaks. Instead, it calculates the BPV for each band $\mathbf{b}_i$ and then integrates the $\text{BPV}(\mathbf{b}_i)$ with the DPC score $\gamma_i$ as a curve to produce a BDPC score, given by $\eta_i = \gamma_i \times \text{BPV}(\mathbf{b}_i)$, which will be used to prioritize all bands for BS similarly to how $\gamma_i$ plays the same role for DPC-BS. Using this BDPC score, two versions of BDPC-BS can be further developed, to be called $b_c$-BDPC and $k$-BDPC. $b_c$-BDPC is developed by fixing the cut-off band distance, $b_c$ at a pre-determined value, while $k$-BDPC is developed by fixing the number of nearest neighbors $k$ at a pre-determined value. Particularly, $b_c$-BDPC extends currently used DPC-BS methods from using two indicators (BD, BLD) to three indicators (BD, BLD, and $\text{BPV}(\mathbf{b}_i)$). On the other hand, $k$-BDPC is new and developed based on a given number of $k$ nearest neighboring bands to calculate the three indictors (BD, BLD, and $\text{BPV}(\mathbf{b}_i)$). Unlike DPC-BS, which requires a pre-specified value for $b_c$, the proposed $b_c$-BDPC can automatically determine the value of $b_c$ by the number of clusters, $n_{\text{clusters}}$. Specifically, for hyperspectral image classification (HSIC), $n_{\text{clusters}}$ can be determined by the number of classes, $n_{\text{classes}}$. As for $k$-BDPC, the value of $k$ can be determined by virtual dimensionality (VD) [27,28].

Finally, to make BDPC more effective for hyperspectral BS, the Euclidean distance, commonly used by DPC, is replaced by a spectral discrimination measure such as spectral angle mapper (SAM) [27], spectral information divergence (SID) [29], or a joint SID with SAM called SIDAM [30], etc., which can be used to calculate BD and also inter-band correlation.

*1.6. Contributions*

Several contributions are summarized as follows:

- A new concept of BPV is introduced into DPC as a third indicator to extend the commonly used two-indicator DPC-BS, (BLD,BD)-DPC-BS, to a new three-indicator BDPC-BS, (BLD,BD,BPV)-BDPC-BS.
- Using (BLD,BD,BPV)-BDPC-BS, two versions of BDPC-BS, $b_c$-BDPC and $k$-BDPC are developed. Specifically, $b_c$-BDPC can be considered as an extension to DPC-BS.
- Automatic rules are particularly derived to determine the value of the cut-off band distance, $b_c$, for $b_c$-BDPC and the number of nearest neighboring bands, $k$, for $k$-BDPC.
- Spectral discrimination measures are used in BDPC-BS to replace Euclidean distance in DPC-BS to better capture BD and spectral inter-band correlation.

The remainder of this paper is organized as follows: Section 2.1 reviews DPC-related works, while Section 2.2 discusses the most recent DPC-BS-based methods. Section 2.3 develops new variants of DPC-BS methods. Section 2.4 introduces the concept of BPV to explore inter-band correlation to extend DPC-BS to BDPC-BS, where two versions of BDPC BS methods, $b_c$-BDPC and $k$-BDPC, are derived in great detail. In particular, the automatic rules for determining the cut-off band distance, $b_c$, and the $k$ nearest neighbors are also derived. Section 3.1 describes three hyperspectral images used for classification experiments.

Section 3.2 conducts extensive experiments for HSIC along with their discussions. Section 4 describes the novelties of this paper and draws some conclusions.

## 2. Methods

DPC was originally proposed by Rodriguez and Laio for data point clustering and has received considerable interest in pattern recognition [26]. In [31] Wei et al. presented an overview on DPC to first analyze the theory of DPC and its performance advantages and disadvantages and then summarizes the improvement of DPC in recent years through the improvement effect via experiments and new ideas for improving DPC algorithm in the future. Most recently, Tobin and Zhang [32] provided theoretical properties that characterize DPC and further develops a clustering algorithm, called Component-wise Peak-Finding (CPF) to deal with detection of erroneous points with high density and large distance to points of higher density as well as incoherent cluster assignment caused by noise.

### 2.1. DPC

The idea of DPC assumes that a cluster center is surrounded by its neighboring data points, with local density measured by the number of its neighbors, and each cluster has a relatively large distance from other clusters. More specifically, DPC designs two indicators to cluster data points. One is local density, defined by:

$$\rho_i = \sum_{j \neq i} \chi_{d_c}(d_c - d_{ij}) \tag{1}$$

where $d_c$ is the predetermined cut-off distance, $d_{ij}$ is the Euclidean distance between two data points $\mathbf{x}_i$ and $\mathbf{x}_j$, and

$$\chi_{d_c}(d_c - d_{ij}) = \begin{cases} 1; & \text{if } d_c - d_{ij} > 0 \\ 0; & \text{otherwise} \end{cases} \tag{2}$$

The other is the cluster distance defined by:

$$\delta_i = \min_{\rho_j > \rho_i} d_{ij} \tag{3}$$

which is calculated by minimizing the distance between data points $\mathbf{x}_i$ and $\mathbf{x}_j$ with local density $\rho_j$ higher than $\rho_i$.

For the data point with the highest local density, $\delta_i^{\max}$, it is defined as:

$$\delta_i^{\max} = \max_j d_{ij} \tag{4}$$

Thus, a point with relatively high $\rho_i$ and large $\delta_i$ can be considered a cluster center. These two quantities are then multiplied together to yield a quantity for a data point $\mathbf{x}_i$, given by:

$$\gamma_i = \rho_i \times \delta_i \tag{5}$$

which can be used to define a DPC score of $\mathbf{x}_i$. Those points that have larger $\gamma_i$ are more likely to be distinguished as cluster centers. With this in mind, (5) can be used to rank the cluster centers.

On many occasions, the discrete value of the local density in (1) is generally replaced by a continuous value, as follows:

$$\rho_i = \sum_{d_{ij} < d_c} \exp\left(-\left(\frac{d_{ij}}{d_c}\right)^\alpha\right) \tag{6}$$

where $\alpha$ is an adjusting factor.

One crucial issue arising from DPC is the determination of $d_c$. Xu et al. [33] introduced density–distance clustering (DDC) to develop an improved automatic density peaks clus-

tering (ADPC) algorithm that could automatically select the suitable cut-off distance, $d_c$, and acquires the optimal number of clusters without additional parameters. Also, DPC generally makes an assumption that cluster centers are often surrounded by data points with lower local density and are far away from other data points with higher local density. However, this assumption is not necessarily true. To address this issue, Wang et al. [34] developed a variational density peak clustering (VDPC) algorithm, which is designed to systematically and autonomously perform the clustering task on datasets with various types of density distributions.

### 2.1.1. *k*-Nearest Neighbors-Based DPC

It is interesting to note that the concept of local density, $\rho_i$ is actually very closely related to *k*-nearest neighbors (*k*NNs). Du et al. [35] utilized *k*NNs to calculate $\rho_i$ to derive DPC-KNN. Since then, many works along this line have been reported [36–39]. Specifically, local density is very closely related to *k*NNs. For each data point $\mathbf{x}_i$, $kNN(\mathbf{x}_i)$ is defined as the set of $k$ data points that are the nearest neighbors close to $\mathbf{x}_i$. Three *k*NN-based DPC methods are of particular interest.

### 2.1.2. DPC-KNN

Due to the fact that the global data structure may lead DPC to miss many clusters, Du et al. [34] proposed a DPC based on *k*NNs (DPC-KNN) by including *k*NNs into DPC to address this issue. Additionally, in high-dimensional data spaces, DPC is likely to generate an incorrect number of clusters. To cope with this problem, DPC-KNN also includes principal component analysis (PCA) into DPC-KNN to derive DPC-KNN-PCA. However, two issues that have significant impacts on DPC performance were not addressed, that is: how to determine the value of $k$ and the number of principal components, both of which were determined empirically. However, for multi-density data scenarios, one parameter cannot satisfy all data sets. So, clustering often cannot achieve good results. To resolve this issue, Yin et al. [40] extended the DPC algorithm to deal with multi-density data. The cut-off distance $d_c$ was selected using KNN to sort the neighbor distances of each data point to draw a line graph of the KNN distance and found the global bifurcation point to divide the data with different densities.

### 2.1.3. G-DPC-KNN

Compared to DPC-KNN, which empirically determined the value of $k$, Jiang et al. [41] developed a method called G-DPC-KNN to calculate the cutoff distance $d_c$ based on the Gini coefficient and then used *k*NNs to find cluster centers. Most recently, Anandarao and Chellasamy [42] also addressed the issue of the random selection of the cut-off distance parameter, $d_c$, by using the Gini index or Gaussian function to make a valid guess of $d_c$. Unfortunately, the same issue arising from DPC-KNN in how to determine an appropriate value of $k$ for G-DPC-KNN remains challenging. Wang et al. [43] developed a novel density peak clustering algorithm for the automatic selection of clustering centers based on K-nearest neighbors (AKDPC) to remedy manual determination of cluster centers and poor performance on complex datasets with varying densities.

### 2.1.4. Shared Nearest Neighbors (SNNs)

Since *k*NNs generally run into an issue where two data points may share certain nearest neighbors (SNNs), it results in double accounts for these shared nearest neighbors. To alleviate this dilemma, shared nearest neighbor (SNN)-based clustering methods were developed. For example, a very early attempt was made by Jarvis and Patrick [44], who introduced a nonparametric clustering algorithm using the concept of similarity based on the sharing of near neighbors. Liu et al. [45] proposed the SNN-based fast search clustering algorithm for finding density peaks, which is based on three newly defined indicators: SNN similarity, local density $\rho$, and distance from the nearest larger density point $\delta$, to take the information from the nearest neighbors and the shared neighbors into account

so as to self-adapt to local surroundings. Specifically, it introduced a two-step allocation method that can accurately recognize and allocate points by counting the number of shared neighbors between the two points and also assign remaining points by finding the clusters to which more neighbors belong. Lv et al. [46] proposed a fast-searching density peak clustering algorithm based on shared nearest neighbors and an adaptive clustering center (DPC-SNNACC) algorithm to automatically determine the number of knee points in the decision graph according to the characteristics of different datasets and the number of clustering centers without human intervention.

### 2.2. DPC-BS

Applications of DPC to BS in hyperspectral imaging have been recently explored. Several recent works related to using DPC to develop BS techniques have emerged in the literature.

First of all, in order for DPC to be applied to BS, we need to interpret each data sample point $\mathbf{x}_i$ as a band vector, $\mathbf{b}_i$. In this case, the data sample set $\{\mathbf{x}_i\}_{i=1}^N$ is replaced with the full band set, $\{\mathbf{b}_l\}_{l=1}^L$, where $\mathbf{b}_l = (b_{l1}, b_{l2}, \cdots, b_{lL})^T$ is the $l^{\text{th}}$ band image and can be considered as a band vector with $L$ spectral band components. So, the sample distance between two data points, $\mathbf{x}_i$ and $\mathbf{x}_j$, $d_{ij} = d(\mathbf{x}_i, \mathbf{x}_j)$ now becomes the band distance (BD) between two bands, $\mathbf{b}_i$ and $\mathbf{b}_j$, $d(\mathbf{b}_i, \mathbf{b}_j)$, which is measured by the Euclidean distance.

$$b_{ij} = d(\mathbf{b}_i, \mathbf{b}_j) = \left\| \mathbf{b}_i - \mathbf{b}_j \right\|_2 = \sum_{l=1}^L \left\| b_{il} - b_{jl} \right\|^2 \tag{7}$$

Using (7), four DPC-based BS methods were recently developed and are described in detail, as follows.

### 2.2.1. Exemplar Component Analysis (ECA)

An early DPC-BS method was developed by Sun et al. [47], who considered BS as exemplar component analysis (ECA) of hyperspectral bands with high local density and large distances from bands with higher densities. An early attempt to extend DPC to hyperspectral BS is exemplar component analysis (ECA) proposed by Sun et al. [47], which defines local density in (6) as:

$$\rho^{\text{ES}}(\mathbf{b}_i) \equiv \rho_i^{\text{ES}} = \sum_{j=1, j \neq i}^L \exp\left( -b_{ij}/2\sigma^2 \right) \tag{8}$$

where $\alpha$ in (6) is set to 1 and $\gamma_i$ in (5), as follows:

$$\gamma^{\text{ES}}(\mathbf{b}_i) \equiv \gamma_i^{\text{ES}} = \rho_i^{\text{ES}} \times \delta_i^{\text{ES}} \tag{9}$$

where ES is the exemplar score, and $\delta^{\text{ES}}(\mathbf{b}_i) \equiv \delta_i^{\text{ES}}$ is defined by (3). The $\gamma_i^{\text{ES}}$ in (9) is then used as a BP criterion to prioritize all bands in descending order and then select the first $n_{\text{BS}}$ largest $\gamma_i^{\text{ES}}$ scores as the desired band subset.

### 2.2.2. Enhanced Fast Density-Peak Clustering (E-FDPC)

To improve ECA's ability to find bands as cluster centers in small regions, Jia et al. [48] developed an enhanced fast density-peak clustering (E-FDPC) method by adopting weights to use DPC scores for BS. A second DPC-BS method is the enhanced fast density-peak clustering (E-FDPC) proposed by Jia et al. [48], which is defined as:

$$s_{ij} = \left\| \mathbf{b}_j - \mathbf{b}_i \right\|_2 = \sum_{l=1}^L \left\| b_{jl} - b_{il} \right\|^2 \tag{10}$$

$$b_{ij}^{\text{E-FDPC}} = \sqrt{s_{ij}}/L \tag{11}$$

$$\rho^{\text{E-FDPC}}(\mathbf{b}_i) \equiv \rho_i^{\text{E-FDPC}} = \sum_{j=1, j \neq i}^L \exp\left( -(b_{ij}^{\text{E-FDPC}}/b_c^{\text{E-FDPC}})^2 \right) \tag{12}$$

and

$$b_c^{\text{E}-\text{FPDC}} = b_{\text{initial}} / \exp(n_{\text{BS}}/L) \tag{13}$$

where $b_{\text{initial}}$ is an initial value of the cut-off threshold, which can be empirically determined as $b_{\text{initial}} = 2\% \times L \times (L-1)^{\text{th}}$ smallest value of $\{s_{ij}\}_{i,j}$. Then, an E-FDPC score is calculated by multiplying the two indicators, $\rho_i^{\text{E}-\text{FPDC}}$ in (12) and $\delta^{\text{E}-\text{FPDC}}(\mathbf{b}_i) \equiv \delta_i^{\text{E}-\text{FPDC}}$ in (13), together as:

$$\gamma^{\text{E}-\text{FPDC}}(\mathbf{b}_i) \equiv \gamma_i^{\text{E}-\text{FPDC}} = \rho_i^{\text{E}-\text{FPDC}} \times \left(\delta_i^{\text{E}-\text{FPDC}}\right)^2 \tag{14}$$

where $\rho_i^{\text{E}-\text{FPDC}}$ and $\delta_i^{\text{E}-\text{FPDC}}$ are determined by $b_{ij}^{\text{E}-\text{FDPC}}$ and $b_c^{\text{E}-\text{FPDC}}$, respectively.

### 2.2.3. Information-Assisted Density Peak Index (IaDPI)

ECA and E-FDPC rank bands by their calculated DPC scores to locate cluster center bands with larger ranking scores. Recently, Luo et al. [49] developed the information-assisted density peak index (IaDPI) method by including band entropy as an additional indicator to further improve E-FDPC. Recently, Luo et al. [49] developed IaDPI to include band entropy as an additional indicator to further improve E-FDPC, where the entropy of a band image $\mathbf{b}_l$ is defined as:

$$H_i = H(\mathbf{b}_i) = -\sum_{l=1}^{L} p_{il} \log p_{il} \tag{15}$$

with $\mathbf{p}_i = (p_{i1}, p_{l2}, \cdots, p_{iL})^T$ defined as the probability vector of the band image $\mathbf{b}_l$ and $p_{il} = \frac{b_{il}}{\sum_{l=1}^{L} b_{il}}$. The local density for IaDPI is then defined as:

$$\begin{aligned} \rho^{\text{IaDPA}}(\mathbf{b}_i) \equiv \rho_i^{\text{IaDPA}} = \\ \sum_{b_{ij}<b_c} \exp\left(-\left(b_{ij}/b_c\right)^{\alpha_1} + (H_i/H_c)^{\alpha_2} + \left(\left(|H_i - H_j|/H_c\right)^{\alpha_2}\right)\right) \end{aligned} \tag{16}$$

where $\alpha_1$ is an adjusting factor set to 2, and $\alpha_2$ is set to 1. Furthermore, it also defines:

$$Tb_{ij} = b_{ij}^{\alpha_1} \cdot H_j^{\alpha_2} \tag{17}$$

to derive:

$$\delta^{\text{IaDPA}}(\mathbf{b}_i) \equiv \delta_i^{\text{IaDPA}} = \sqrt[(\alpha_1+\alpha_2)]{Tb_{ij}^{\alpha_1} \cdot H_j^{\alpha_2}} \tag{18}$$

where $j$ is determined by:

$$j = \arg\left\{\min_{j:\rho_j>\rho_i} Tb_{ij}\right\} \tag{19}$$

Finally, the IaDPA score is calculated by:

$$\gamma^{\text{IaDPA}}(\mathbf{b}_i) \equiv \gamma_i^{\text{IaDPA}} = \rho_i^{\text{IaDPA}} \times \delta_i^{\text{IaDPA}} \tag{20}$$

### 2.2.4. Shared Nearest Neighbors Network (SNNC)

Because $kNN(\mathbf{b}_i)$ and $kNN(\mathbf{b}_j)$ may share the same nearest neighbors, Li et al. [50] proposed an efficient SNN clustering (SNNC) method for hyperspectral optimal BS, which can obtain the local density of each band using SNN to more accurately reflect the local distribution characteristics. In other words, the local density of each band obtained by SNN can more accurately reflect the similarities in the local distribution characteristics, measured by:

$$SNN(\mathbf{b}_i, \mathbf{b}_j) = |kNN(\mathbf{b}_i) \cap kNN(\mathbf{b}_j)| \tag{21}$$

From (21), an SNN-based local density can now be defined as:

$$\rho^{k-\text{SNNC}}(\mathbf{b}_i) \equiv \rho_i^{k-\text{SNNC}} = \sum_{\mathbf{b}_j \in kNN(\mathbf{b}_i)} \exp\left(-\frac{d(\mathbf{b}_i, \mathbf{b}_j)}{SNN(\mathbf{b}_i, \mathbf{b}_j)+1}\right) \tag{22}$$

where $d(\mathbf{b}_i, \mathbf{b}_j) = \sum_{l=1}^{L} (b_{il} - b_{jl})^2$.

$$\delta^{k-\text{SNNC}}(\mathbf{b}_i) \equiv \delta_i^{k-\text{SNNC}} = \begin{cases} \min_j d(\mathbf{b}_i, \mathbf{b}_j) \text{ if } \rho_i^{k-\text{SNNC}} < \rho_j^{k-\text{SNNC}} \\ \max_j d(\mathbf{b}_i, \mathbf{b}_j) \text{ otherwise} \end{cases} \quad (23)$$

In addition, to acquire a band subset containing a large amount of information, the information entropy was proposed to be used as one of the weight factors. More specifically, an entropy-based BP criterion for SNNC is derived by combining (22) and (23) with the information entropy of band $\mathbf{b}_i$ defined in (15), referred to as $H^{k-\text{SNNC}}(\mathbf{b}_i) \equiv H_i^{k-\text{SNNC}}$, as follows:

$$\gamma^{k-\text{SNNC}}(\mathbf{b}_i) \equiv \gamma_i^{k-\text{SNN}} = \rho_i^{k-\text{SNNC}} \times \delta_i^{k-\text{SNNC}} \times H_i^{k-\text{SNNC}} \quad (24)$$

which corresponds to $\gamma_i$ in (5).

Finally, SNNC also developed a method for automatically selecting the optimal band subset, designed based on the slope change.

### 2.3. New Variations of DPC-BS

While DPC has shown potential and promise in BS for hyperspectral BS, there are several issues that remain unsolved. The first issue is the use of the Euclidean distance, which may not effectively capture spectral correlation compared to the use of a spectral discrimination measure. The second issue is the lack of a mechanism to address inter-band correlation. It should be noted that although IaDPI and SNNC use band information entropy as an additional indicator, it only measures the uncertainty of each band without considering inter-band spectral correlation. The third issue is determining the value of the cut-off BD, $b_c$, or the number of nearest neighboring band images, $k$. This is because DPC-BS performance is heavily determined by its BLD, $\rho_i$ and BD, $\delta_i$, which are actually calculated using either $b_c$ or $k$. Unfortunately, these two parameters, $b_c$ and $k$, are indeed inter-correlated and cannot be determined independently. In other words, one parameter determines the other. To address this issue, we further derive two versions of DPC-BS, namely $b_c$-DPC-BS and $k$-DPC-BSD PC-BS, by manipulating $b_c$ or $k$.

### 2.3.1. $b_c$-DPC-BS

The $b_c$-DPC-BS to be developed in this section is similar to ECA, E-FDPC, and IaDPA in the sense that the cut-off band distance $b_c$ is determined and fixed at a certain value a priori. It can be considered as a variant of DPC-BS wherein the Euclidean distance used to calculate the BLD, $\rho_i$ and BD, $\delta_i$ is replaced with a spectral discrimination measure, $m$. In this case, $d_c$ is replaced by $b_c$ and $\rho_i$ in (1) becomes:

$$\rho_i^{\text{BLD}} = \sum_{j \neq i} \chi_{b_c}(b_c - b_{ij}) \quad (25)$$

where $b_c$ is the predetermined band distance (BD) threshold, $b_{ij}$ is the spectral distance between $\mathbf{b}_i$ and $\mathbf{b}_j$, defined as:

$$b_{ij} = m(\mathbf{b}_i, \mathbf{b}_j) \quad (26)$$

where $m$ can be any type of spectral measure, such as SAM, SID, or SIDAM. In this case, Equations (2) and (3) are modified and re-defined as:

$$\chi_{b_c}(b_c - b_{ij}) = \begin{cases} 1; \text{ if } b_c - b_{ij} < 0 \\ 0; \text{ otherwise} \end{cases} \quad (27)$$

$$\delta^{b_c-\text{BD}}(\mathbf{b}_i) \equiv \delta_i^{b_c-\text{BD}} = \min_{\rho_j^{\text{BLD}} > \rho_i^{\text{BLD}}} b_{ij} \quad (28)$$

For the band with the highest band local density, $\delta_i^{\max-\mathrm{BD}}$ in (5) is re-defined as:

$$\delta^{\max-\mathrm{BD}}(\mathbf{b}_i) \equiv \delta_i^{\max-\mathrm{BD}} = \max_j b_{ij} \tag{29}$$

Analogous to (6), (25) can also be modified as kernel-based BLD. $\rho_i^{b_c-\mathrm{BLD}}$ is defined as:

$$\rho^{b_c-\mathrm{BLD}}(\mathbf{b}_i) \equiv \rho_i^{b_c-\mathrm{BLD}} = \sum_{j=1,\neq i}^{L} \exp\left(-\left(\frac{b_{ij}}{b_c}\right)^{\alpha}\right) \tag{30}$$

where $\alpha$ is set to 2 for our experiments. Combining (30) with (28), we can define the $b_c$-DPC score as:

$$\gamma^{b_c-\mathrm{DPC}}(\mathbf{b}_i) \equiv \gamma_i^{b_c-\mathrm{DPC}} = \rho_i^{b_c-\mathrm{BLD}} \times \delta_i^{b_c-\mathrm{BD}} \tag{31}$$

which corresponds to $\gamma_i$ in (5).

### 2.3.2. k-DPC-BS

The $k$-DPC-BS presented in this subsection is similar to DPC based on $k$-nearest neighboring bands. But, it introduces a new aspect by interchanging the roles played by BLD and BD in $b_c$-DPC-BS compared to $k$-DPC-BS. Specifically, it assumes that the value of $k$ is yielded a priori to calculate the BD of $\mathbf{b}_i$, $\delta_i^{k-\mathrm{BD}}$, which embraces $k$ bands surrounding $\mathbf{b}_i$. Then, $\rho_i^{b_c-\mathrm{BLD}}$ can be further calculated by $\delta_i^{k-\mathrm{BD}}$. It is a sequential process of $k \to \delta_i^{k-\mathrm{BD}} \to \rho_i^{b_c-\mathrm{BLD}}$. In this case, we consider a cluster centered at a band vector $\mathbf{b}_i$ and expand its range until the cluster contains $k$ bands surrounding $\mathbf{b}_i$ according to their spectral similarities.

For each $\mathbf{b}_i$, let the set contain the $k$ nearest neighboring bands surrounding $\mathbf{b}_i$, denoted by $\mathbf{B}_i^k$. Then, we can define a criterion similar to $\rho_i$ in (1) as:

$$\rho^{k-\mathrm{BLD}}(\mathbf{b}_i) \equiv \rho_i^{k-\mathrm{BLD}} = \max_{\mathbf{b}_j \in \mathbf{B}_i^k} m(\mathbf{b}_i, \mathbf{b}_j) \tag{32}$$

The minimum distance of $\mathbf{b}_i$ from all other bands $\mathbf{b}_j$ with $\rho_j^{k-\mathrm{BLD}}$ greater than $\rho_i^{k-\mathrm{BLD}}$, denoted by $\delta_i^{k-\mathrm{BD}}$ is defined as:

$$\delta^{k-\mathrm{BD}}(\mathbf{b}_i) \equiv \delta_i^{k-\mathrm{BD}} = \min_{\rho_j^{k-\mathrm{BLD}} > \rho_i^{k-\mathrm{BLD}}} m(\mathbf{b}_i, \mathbf{b}_j) \tag{33}$$

Finally, a $k$-DPC score, denoted by $\gamma^{k-\mathrm{DPC}}(\mathbf{b}_i)$, can be defined by $\gamma_i^{k-\mathrm{DPC}}$ in correspondence to $\gamma_i$ in (5) as:

$$\gamma^{k-\mathrm{DPC}}(\mathbf{b}_i) \equiv \gamma_i^{k-\mathrm{DPC}} = \rho_i^{k-\mathrm{BLD}} \times \delta_i^{k-\mathrm{BD}} \tag{34}$$

### 2.4. BDPC-BS

According to (31) and (34), the $b_c$-DPC-BS and $k$-DPC-BS scores do not take into account the inter-band correlation, and neither do ECA, E-FDPC, IaDPA, or $k$-SNNC. So, to address this issue, a new concept derived from prominent band peaks proposed in [5], called band prominence value (BPV), BPV($\mathbf{b}_i$) for each band vector $\mathbf{b}_i$ is introduced into DPC as a third indicator. By incorporating BPV into BLD, $\rho_i$, and BD, $\delta_i$, we can further extend the two-indicator (BLD,BD)-DPC-BS and $b_c$-DPC-BS and $k$-DPC-BS in Section 2.3 to three indicator (BLD,BD,BPV)-DPC-BS, $b_c$-BDPC-BS, and $k$-BDPC-BS. This is similar to SNNC, which includes a third indicator, information entropy $H_i^{k-\mathrm{SNNC}}$ in (24).

The concept of BPV can be applied to any DPC method. It plots DPC scores obtained by (9) for ECA, (14) for E-FDPC, (20) for IaDPA, (24) for $k$-SNNC, (31) for $b_c$-DPC-BS, and (34) for $k$-DPC-BS, etc., as a curve where each point on the curve corresponds to the band prominence value (BPV) of one band. For example, for band C, BPV(C) is calculated and illustrated in Figure 1, which shows that the band prominence value of the peak C can be calculated.

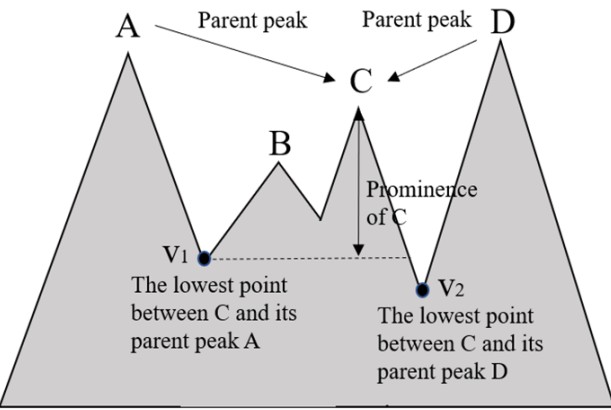

**Figure 1.** The vertical line represents the prominence of peak C, which is the difference between the height of C and the height of $v_1$.

To better understand how BPV is calculated according to Figure 1, a detailed step-by-step implementation for calculating BPV is summarized in Algorithm 1. As a matter of fact, how to calculate the prominence of a peak is available in Matlab and can be performed by findpeaks on https://www.mathworks.com/help/signal/ug/prominence.html (accessed on 5 March 2024).

---

**Algorithm 1:** Algorithm for Calculating Prominence Scores of Bands

---

1. **Input**: Let the local minimum (LM) points set be $\boldsymbol{\Omega}_{\mathrm{LM}} = \left\{ \mathbf{b}_{j_l} \right\}_{l=1}^{n_{\mathrm{LM}}}$ and $\tilde{\boldsymbol{\Omega}} = \boldsymbol{\Omega} - \boldsymbol{\Omega}_{\mathrm{LM}}$, where $\boldsymbol{\Omega}$ is the full bands set.

2. For $\mathbf{b}_i \in \tilde{\boldsymbol{\Omega}}$, calculate the prominence by the following procedure:

    a. Extend a horizontal line from point $\mathbf{b}_i$ to the left and the right until the line either

       • crosses the curve because there is a higher point

       or

       • reaches the left or right end of the curve.

       Then, the left interval and right interval of $\mathbf{b}_i$ are defined.

    b. Let $\gamma_i$ be either $\gamma_i^{b_c - \mathrm{BDPC}}$ or $\gamma_i^{k - \mathrm{BDPC}}$. We can find the BPV of $\mathbf{b}_i$, BPV($\mathbf{b}_i$), as follows:

       i   Find the minimum point $\mathbf{b}_l$ in the left interval. Let $\varsigma_l = \gamma_l$ if $\gamma_l < \gamma_i$. Otherwise, let $\varsigma_l = 0$.

       ii  Find the minimum point $\mathbf{b}_j$ in the right interval. Let $\varsigma_j = \gamma_j$ if $\gamma_j < \gamma_i$. Otherwise, let $\varsigma_j = 0$.

       iii **Output** BPV($\mathbf{b}_i$) $= \gamma_i - \max(\varsigma_l, \varsigma_j)$

---

Once $\{\mathrm{BPV}(\mathbf{b}_i)\}_{i=1}^{L}$ for all band vectors $\{\mathbf{b}_i\}_{i=1}^{L}$ are calculated by the above algorithm, we can multiply BPV($\mathbf{b}_i$) with $\gamma_i^{b_c - \mathrm{BDPC}}$ in (31) or $\gamma_i^{k - \mathrm{BDPC}}$ in (34) to yield new BPV-based BP criteria for $b_c$-BDPC-BS, defined as:

$$\eta_i^{b_c - \mathrm{BDPC}} = \gamma_i^{b_c - \mathrm{BDPC}} \times \mathrm{BPV}(\mathbf{b}_i) \tag{35}$$

and for $k$-BDPC BS, defined as:

$$\eta_i^{k - \mathrm{BDPC}} = \gamma_i^{k - \mathrm{BDPC}} \times \mathrm{BPV}(\mathbf{b}_i) \tag{36}$$

Assuming $n_{\mathrm{BS}}$ is the number of bands to be selected, $b_c$-BDPC and $k$-BDPC can use (35) and (36) to select the bands with the first $n_{\mathrm{BS}}$ largest values of $\eta_i^{b_c - \mathrm{BDPC}}$ and $\eta_i^{k - \mathrm{BDPC}}$ as desired bands for BS, respectively.

A final remark is noteworthy. The key difference between the idea used in BDPC-BS and SMI-BS in [5] is that BDPC-BS calculates BPV for each band as an indicator jointly used with BLD and BD for BS, compared to SMI-BS, which find prominent peaks for BS.

### 2.4.1. Determination of $b_c$ and $k$

Now, we come to the last issue, which is how to determine the values of the two parameters, $b_c$ and $k$, which have significant impacts on the performance of DPC-BS and BDPC-BS methods. Unfortunately, these two parameters were generally determined a priori or empirically in the past. This section discusses several approaches to determine appropriate values of $b_c$ and $k$ for BDPC automatically.

### 2.4.2. Determination of $b_c$

As noted above, when DPC is applied to BS, the total number of data samples, $N$, is now replaced by $L$, the total number of bands. In [35], $b_c$ is determined by:

$$b_c = b_{\lceil L_d \times p\%/100 \rceil} \tag{37}$$

where the $b_{\lceil L_d \times p\%/100 \rceil}{}^{\text{th}}$ distance is selected according to an ascending order among all possible distances between two band vectors. In (37), $\lceil x \rceil$ is the upper ceiling of $x$, defined by the smallest integer equal to or greater than $x$, and $L_d = L(L-1)/2$ is the total number of distances, and $p\%$ is a pre-specified percentage.

As an alternative to (37), IaDPI [49] selects:

$$b_c = Vd(\lceil 2\% \times L \times (L-1) \rceil) / \exp(k/L) \tag{38}$$

where $Vd(x)$ is the $x^{\text{th}}$ smallest distance among all possible distances between two data samples, with $L$ being the total number of bands.

In [41], it assumed that the potential energy in the data domain was similar to the local density of the points in the dataset. As a result, the potential energy of each point can be used as an indicator of the overall distribution of the dataset to estimate the potential energy of the whole dataset. In this case, we can define the potential energy of each band image $\mathbf{b}_i$ as:

$$\delta_i(\sigma) = \sum_{j=1, ji}^{L} \exp\left( -\left( ||\mathbf{b}_j - \mathbf{b}_i|| / \sigma \right)^2 \right) \tag{39}$$

where $\sigma$ plays a similar role as $d_c$ does in (1). So, calculating the potential energy of band sets is equivalent to calculating the local density of band sets. In doing so, it introduces the Gini coefficient, denoted by:

$$G(\sigma) = 1 - \sum_{i=1}^{L} (\delta_i(\sigma)/Z) \tag{40}$$

where:

$$Z = \sum_{i=1}^{L} \delta_i(\sigma) \tag{41}$$

is the total potential energy of all band images. We then define:

$$\sigma_{\text{opt}} = \min_{\sigma} G(\sigma) \tag{42}$$

which determines the desired $b_c$ as:

$$b_c = \sigma_{\text{opt}} \tag{43}$$

Interestingly, (37), (38) and (43), which are used to determine $b_c$, have nothing to do with the number of clusters, $n_{\text{clusters}}$, which is supposed to be the key parameter of BDPC. To address this issue, we developed an automatic algorithm to determine $n_{\text{clusters}}$-based $b_c$, which is very easy and simple to implement. Its idea is to first use the well-known K-means clustering method, also known as ISODATA [51], to group a dataset into $n_{\text{clusters}}$,

$\{C_j\}_{j=1}^{n_{\text{clusters}}}$. Then, for each cluster $C_j$, we calculated its cluster center, $\{\boldsymbol{\mu}_j\}_{j=1}^{n_{\text{clusters}}}$, by its mean. Finally, we found $b_c$ $b_c = \min_{j, \mathbf{r} \in C_j} m(\mathbf{r}, \boldsymbol{\mu}_j)$.

Despite the fact that ISODATA and K-means methods are essentially the same algorithm, ISODATA is used to avoid confusion between K-means and *k*NNs because the uppercase "K" used by the K-means method is the number of clusters, while the lowercase italic "*k*" used by *k*NNs is the number of nearest neighbors.

Unlike DPC, which determines $d_c$ empirically, $b_c$-BDPC automatically determines $b_c$ by the number of clusters, $n_{\text{clusters}}$. So, if we assume that each class is specified by one cluster, then $b_c$ is in turn determined by the number of classes, $n_{\text{classes}}$ of interest. Once $b_c$ is determined, $\rho_i^{b_c-\text{BLD}}$ can be determined accordingly, i.e., $b_c \to \rho_i^{b_c-\text{BLD}}$.

### 2.4.3. Determination of *k*

As an alternative to using $b_c$ to determine the value of $k$, we can directly use the value of $k$ to calculate local density and BD without appealing to $b_c$. For example, $k$ can be selected as:

$$k = \frac{p\%}{100} \times L \qquad (44)$$

with $p\%$ predetermined empirically. Such selection is a trial-and-error and not practical.

To resolve this issue, virtual dimensionality (VD) introduced in [27,28] can be used for this purpose. It is first used to estimate the number of bands to be selected, and then $k$ can be determined by:

$$k_{\text{BS}} = 2 \times \left\lceil \frac{L}{n_{\text{BS}}} \right\rceil \qquad (45)$$

where "2" is included to take the adjacent bands on both sides.

### 2.4.4. Determination of $n_{\text{BS}}$

It should be noted that in addition to VD in (45) to determine the value of $k$, [50] also developed an automatic rule for selecting an optimal band subset using the slope change, defined as follows:

$$p = \{i | |k_i| - |k_{i+1}| \geq \gamma\} \text{ for } i = 1, 2, \cdots, L - 2 \qquad (46)$$

where:

$$q = \sum_{i=1}^{L-1} ||k_i| - |k_{i+1}|| \text{ and } \gamma = q/(L-2) \qquad (47)$$

$$n_{\text{BS}} = \max\{p\} \qquad (48)$$

## 3. Experiments

### 3.1. Images to Be Used for Experiments

Three popular hyperspectral images that have been studied extensively for HSIC were used for experiments.

### 3.1.1. Purdue Indiana Indian Pines

The first image is the Purdue Indiana Indian Pines test site with an aerial view, shown in Figure 2a, along with its ground truth of 17 class maps in Figure 2b and their class labels in Figure 2c. Table 1 tabulates the number of data samples in each class, where there are four small classes with less than 10 samples: classes 7, 9, 1, 16, and three classes with more than 1000 samples: classes 14, 2, and 11. So, this scene clearly has an imbalanced class issue in classification. It is an airborne visible/infrared imaging spectrometer (AVIRIS) image scene and has a size of $145 \times 145 \times 220$ pixel vectors, with water absorption bands (bands 104–108 and 150–163, 220). So, a total of 220 bands were used for experiments. It should be noted that in many reports, 200 bands were used by excluding water absorption bands. However, for BS, it is believed that a full set of 220 bands should be used for band integrity.

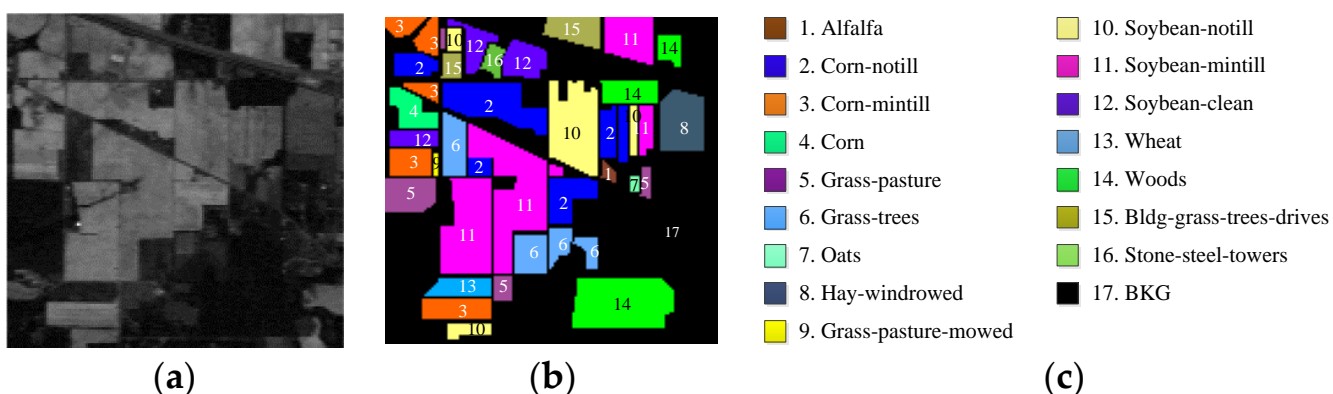

**Figure 2.** AVIRIS image scene: Purdue Indiana Indian Pines test site. (**a**) Band 186 (2162.56 nm). (**b**) Ground truth map. (**c**) Classes by colors.

**Table 1.** Class labels of Purdue Indiana Indian Pines with number of data samples in each class.

| | | | | | |
|---|---|---|---|---|---|
| class 1 (46) | Alfalfa | class 7 (28) | grass/pasture-mowed | class 13 (205) | wheat |
| class 2 (1428) | corn-notill | class 8 (478) | hay-windrowed | class 14 (1265) | woods |
| class 3 (830) | corn-min | class 9 (20) | oats | class 15 (386) | bldg-grass green-drives |
| class 4 (237) | corn | class 10 (972) | soybeans-notill | class 16 (93) | stone-steel towers |
| class 5 (483) | grass/pasture | class 11 (2455) | soybeans-min | class 17 (10,249) | BKG |
| class 6 (730) | grass/trees | class 12 (593) | soybeans-clean | | |

### 3.1.2. Salinas

A second image dataset is Salinas in Figure 3a, which is also an AVIRIS scene. It was collected over Salinas Valley, California, with a spatial resolution of 3.7 m per pixel with a spectral resolution of 10 nm. It has a size of $512 \times 217 \times 224$, with 20 water absorption bands, 108–112, 154–167, and 224. So, a total of 224 bands were used for experiments. Figure 3b,c shows the color composite of the Salinas image along with the corresponding ground truth class labels. Unlike the Purdue data, the Salinas scene does not have the issue of imbalanced classes, as tabulated in Table 2, where the smallest class is class 13 with 916 data samples.

**Table 2.** Class labels of Salinas with number of data samples in each class.

| | | | |
|---|---|---|---|
| class 1 (2009) | Brocoli_green_weeds_1 | class 10 (3278) | Corn_senesced_green_weeds |
| class 2 (3726) | Brocoli_green_weeds_2 | class 11 (1068) | Lettuce_romaine_4wk |
| class 3 (1976) | Fallow | class 12 (1927) | Lettuce_romaine_5wk |
| class 4 (1394) | Fallow_rough_plow | class 13 (916) | Lettuce_romaine_6wk |
| class 5 (2678) | Fallow_smooth | class 14 (1070) | Lettuce_romaine_7wk |
| class 6 (3959) | Stubble | Class 15 (7268) | Vinyard_untrained |
| class 7 (3579) | Celery | class 16 (1807) | Vinyard_vertical_trellis |
| class 8 (11,271) | Grapes_untrained | class 17 (56,975) | BKG |
| class 9 (6203) | Soil_vinyard_develop | | |

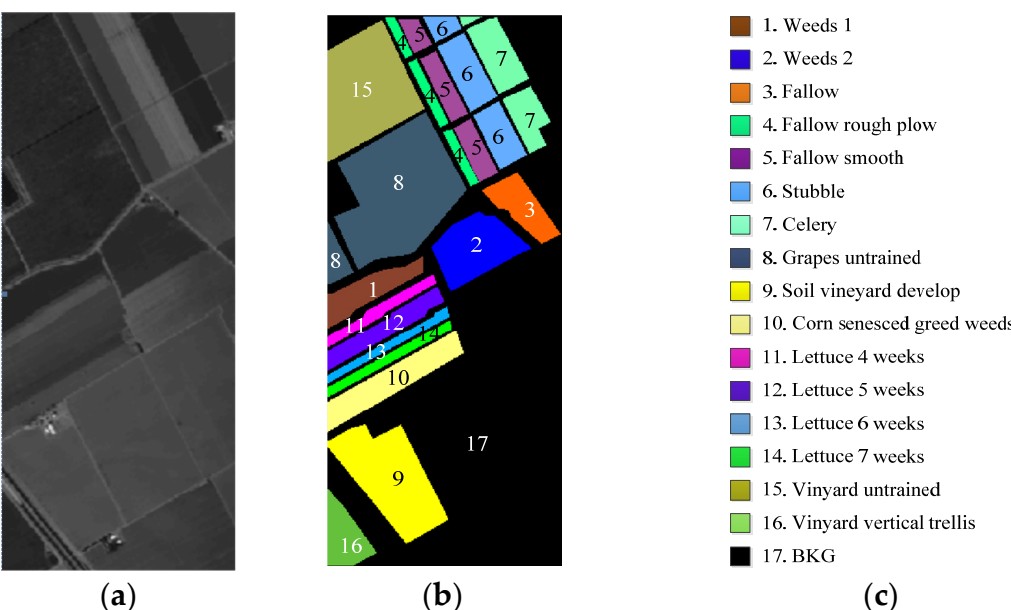

**Figure 3.** Ground-truth of Salinas scene with 16 classes. (**a**) Salinas scene. (**b**) Ground-truth image. (**c**) Classes by colors.

### 3.1.3. University of Pavia

A third hyperspectral image dataset used for experiments was the University of Pavia image shown in Figure 4, which is an urban area surrounding the University of Pavia, Italy. It was recorded by the ROSIS-03 satellite sensor over an urban area surrounding the University of Pavia, Italy. It has a size of $610 \times 340 \times 115$ with a spatial resolution of 1.3 m per pixel and a spectral coverage ranging from 0.43 to 0.86 μm with a spectral resolution of 4 nm (the 12 most noisy channels were removed before experiments). Figure 4b provides its ground-truth map of nine classes along with color class labels in Figure 4c. Table 3 also tabulates the number of data samples in parentheses collected for each class. Like the Salinas scene, this scene also has very large classes, with only one small class having less than 1000 data samples: class 9 with 947 samples. However, this scene has a more complicated BKG than the other two studied scenes, as already shown in [15,16], where the precision rate of this scene was much lower than the other two scenes.

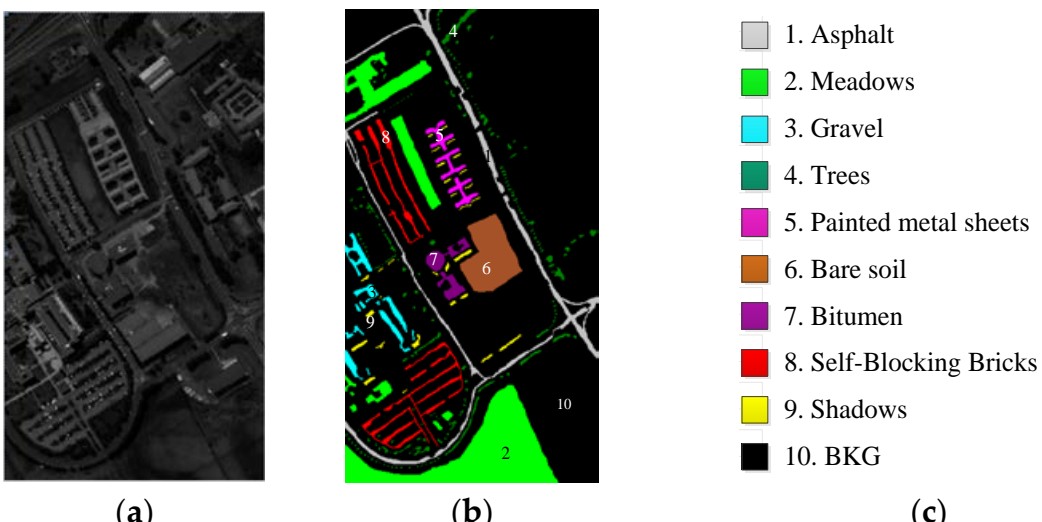

**Figure 4.** Ground-truth of University of Pavia scene with nine classes. (**a**) University of Pavia scene. (**b**) Ground-truth map. (**c**) Classes by colors.

**Table 3.** Class labels of University of Pavia with number of data samples in each class.

| class 1 (6631) | Asphalt | class 5 (1345) | Painted metal sheets | Class 9 (947) | Shadows |
|---|---|---|---|---|---|
| class 2 (18,649) | Meadows | class 6 (5029) | Bare Soil | Class 10 (164,624) | BKG |
| class 3 (2099) | Gravel | class 7 (1330) | Bitumen | | |
| class 4 (3064) | Trees | class 8 (3682) | Self-Blocking Bricks | | |

*3.2. Experimental Results and Discussions*

In order to conduct credible experiments for a comparative analysis, three crucial elements are considered and investigated:

(a)  Number of bands needed to be selected, $n_{BS}$.

Since BS is unsupervised, $n_{BS}$ must be determined without prior knowledge. In this case, VD was used for this purpose.

(b)  Classifiers used for HSIC.

There are many classifiers available and reported in the literature. Selecting a good candidate to be used for HSIC will be a challenge. In this paper, its selection is based on three aspects. One is that it should be effective and perform reasonably well compared to many existing state-of-the-art classifiers. Another is that it should be simple enough to be implemented so that those who are interested in our work can repeat experiments. A third one is that it does not require too many parameters to be tuned, unlike many deep learning-based HSIC methods. It turns out that spectral-spatial classifiers meet these requirements and are preferred to deep learning-based classifiers when it comes to experiments. This is because $b_c$-BDPC and $k$-BDPC methods are steady with the parameters $b_c$ and $k$ being determined automatically. In this case, spectral-spatial classifiers are more appropriate to be used as classifiers than deep learning-based classifiers, which require a number of model parameters to be tuned empirically. Specifically, the Iterative EPF (IEPF) method in [52] was selected as the classifier for experiments because it has been shown to significantly improve EPF-based methods [53]. There were four IEPF methods proposed in [52] that performed similarly. The IEPF-G-g was chosen as its representative for our comparison because IEFP-G-g used guided filters, as opposed to IEPF-B-g, which used bilateral filters [5]. The IEPF-G-g was implemented with the selection of training samples for each class, which followed the same procedure that was used in [52]. Table 4 tabulates the number of training samples selected for each class across the three datasets, and the remaining data samples were used as test data samples.

**Table 4.** Number of training samples of each class for three datasets.

| Class | 1 | 2 | 3 | 4 | 5 | 6 | 7 | 8 | 9 | 10 | 11 | 12 | 13 | 14 | 15 | 16 | Total |
|---|---|---|---|---|---|---|---|---|---|---|---|---|---|---|---|---|---|
| Purdue | 25 | 91 | 80 | 65 | 68 | 73 | 14 | 70 | 11 | 81 | 111 | 70 | 66 | 84 | 69 | 47 | 1025 |
| Salinas | 67 | 67 | 67 | 67 | 68 | 67 | 69 | 69 | 67 | 69 | 67 | 67 | 67 | 68 | 68 | 69 | 1083 |
| U. of Pavia | 100 | 100 | 100 | 100 | 100 | 100 | 100 | 100 | 100 | | | | | | | | 900 |

(c)  BS methods selected for comparison

There are two classes of BS methods of interest. One comprises DPC-based BS methods: ECA, E-FDPC, IaDPI, DPC-KNN, G-DPC-KNN, and Shared Nearest Neighbor Network (SNNC) [50]. The other class is made up of the most recently developed BS methods, such as sequential CCBSS (SQ-CCBSS) [14], successive CCBSS (SC-CCBSS) [14], ref. [8], MDPP [11], DSEBS [12], Linearly Constrained Minimum Variance Sequential Feedforward Band Selection (LCMV-SFBS) [15], Linearly Constrained Minimum Variance Sequential Backward Band Selection (LCMV-SBBS) [15], Feedforward Class Signature-Constrained Band Prioritization (FCSCBS-BP) [16], and Backward Class Signature-Constrained Band Prioritization (BCSCBS-BP) [16]. Since Self-Mutual Information (SMI)-BS was shown in [5]

to perform better than the above-mentioned BS methods, it is sufficient to choose SMI-BS for comparison.

With all things considered, this section compares six DPC-based methods, $b_c$-BDPC and $k$-BDPC, against ECA, E-FDPC, IaDPI, DPC-KNN, G-DPC-KNN, SNNC, and BP-based SMI-BS using three spectral discrimination measures: SAM, SID, and SIDAM. Table 5 lists the parameter setting used for all compared BS methods along with the computer environment: Intel® Core(TM) i7-9750H CPU 2.6 GHz with RAM: 16 G.

**Table 5.** Parameter setting for compared DPC-BS methods.

| DPC Methods | $k$ | $b_c$ |
|:---:|:---:|:---:|
| $b_c$-BDPC | | $n_{\text{clusters}}$ |
| IaDPI | | $2\% \times L \times (L-1)/\exp(n_{\text{BS}}/L)$ |
| $k$-BDPC | $k_{\text{BS}} = 2 \times \lceil L/n_{\text{BS}} \rceil$ | |
| DPC-kNN | $k_{NN} = p\% \times N$ | |
| G-DPC-kNN | emperical selection | Gini coefficient |
| SNNC | $k = 3$ | |

### 3.2.1. Purdue Indian Pines

According to [15,16], VD was estimated to be 18. Figures 5 and 6 plot $\eta_i^{b_c-\text{BDPC}}$ in (39) and $\eta_i^{k-\text{BDPC}}$ in (40), which were produced by $b_c$-BDPC and $k$-BDPC using (a) SID, (b) SAM, and (c) SIDAM for the Purdue Indian Pines scene, respectively, where the red dots are 18 selected bands according to the peaks of BPV, as tabulated in Table 6. In addition, Table 6 also includes 18 bands selected by SMI-BS, ECA, E-FDPC, IaDPI, DPC-kNN, G-DPC-kNN, SNNC, and Uniform BS, where parameters used by various methods are specified in Table 5.

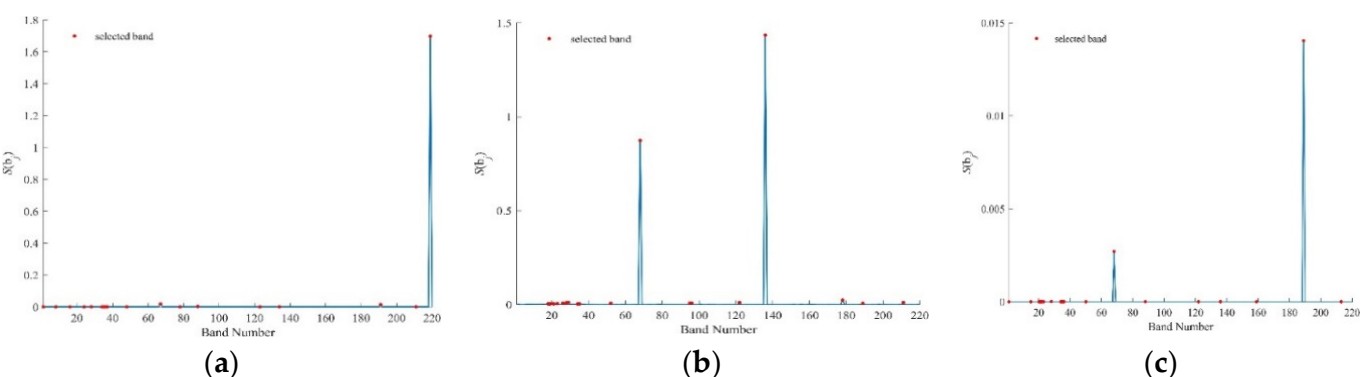

**Figure 5.** $\eta_i^{b_c-\text{BDPC}}$ values by $b_c$-BDPC for the Purdue data using (**a**) SID, (**b**) SAM, (**c**) SIDAM.

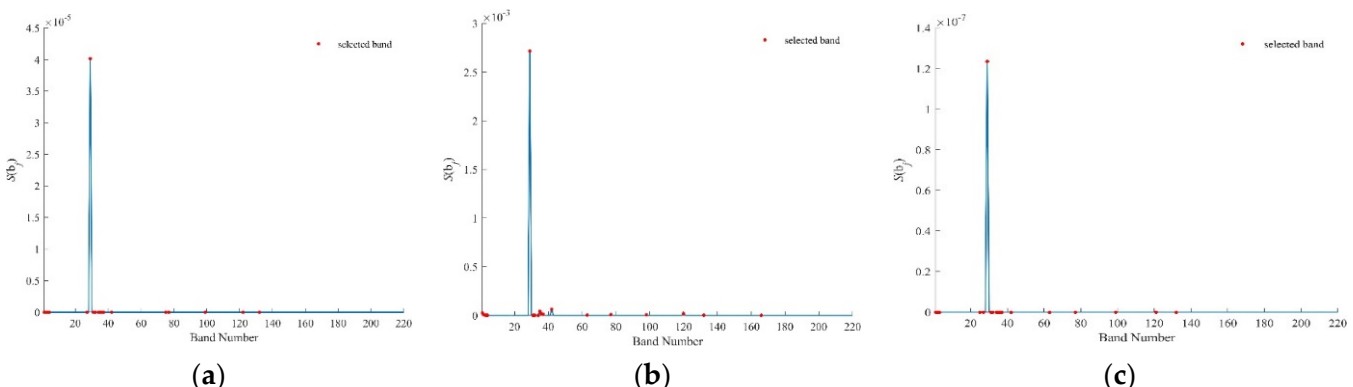

**Figure 6.** $\eta_i^{k-\text{BDPC}}$ values by $k$-BDPC for the Purdue data using (**a**) SID, (**b**) SAM, (**c**) SIDAM.

**Table 6.** 18 bands selected for the Purdue data.

| | Selected Bands ($n_{VD}$ = 18) | | | | | | | | | | | | | | | | | |
|---|---|---|---|---|---|---|---|---|---|---|---|---|---|---|---|---|---|---|
| $b_c$-BDPC-SID | 219 | 67 | 191 | 88 | 28 | 134 | 48 | 211 | 16 | 24 | 1 | 35 | 123 | 34 | 36 | 78 | 37 | 8 |
| $b_c$-BDPC-SAM | 136 | 68 | 178 | 29 | 122 | 211 | 28 | 96 | 95 | 26 | 189 | 52 | 23 | 21 | 35 | 18 | 19 | 34 |
| $b_c$-BDPC-SIDAM | 189 | 68 | 28 | 88 | 136 | 159 | 15 | 50 | 21 | 213 | 22 | 122 | 1 | 20 | 35 | 34 | 23 | 36 |
| $k$-BDPC-SID | 29 | 35 | 42 | 1 | 36 | 122 | 37 | 77 | 99 | 2 | 32 | 75 | 34 | 132 | 3 | 31 | 27 | 4 |
| $k$-BDPC-SAM | 29 | 42 | 35 | 1 | 36 | 120 | 37 | 77 | 98 | 2 | 63 | 132 | 3 | 32 | 166 | 4 | 31 | 34 |
| $k$-BDPC-SIDAM | 29 | 35 | 42 | 1 | 36 | 121 | 37 | 77 | 99 | 34 | 2 | 32 | 63 | 31 | 3 | 132 | 27 | 25 |
| SMI-BS | 59 | 117 | 125 | 38 | 40 | 180 | 173 | 20 | 168 | 31 | 34 | 12 | 114 | 192 | 160 | 108 | 95 | 92 |
| ECA | 164 | 129 | 67 | 83 | 193 | 52 | 14 | 197 | 28 | 110 | 147 | 103 | 112 | 111 | 149 | 208 | 165 | 107 |
| E-FDPC | 194 | 32 | 2 | 100 | 98 | 61 | 1 | 36 | 35 | 99 | 60 | 31 | 101 | 76 | 62 | 13 | 58 | 59 |
| IaDPI | 171 | 115 | 8 | 208 | 87 | 58 | 142 | 46 | 77 | 135 | 31 | 176 | 97 | 190 | 15 | 92 | 54 | 160 |
| DPC-KNN ($k_{BS}$) | 150 | 136 | 187 | 146 | 147 | 119 | 145 | 116 | 103 | 110 | 114 | 101 | 112 | 115 | 125 | 111 | 205 | 208 |
| DPC-KNN ($k$ = 2%L) | 37 | 36 | 2 | 75 | 46 | 1 | 57 | 61 | 42 | 4 | 7 | 100 | 32 | 3 | 47 | 35 | 62 | 183 |
| G-DPC-KNN ($k_{BS}$) | 204 | 206 | 203 | 205 | 148 | 202 | 112 | 207 | 209 | 167 | 201 | 65 | 69 | 70 | 126 | 129 | 132 | 133 |
| G-DPC-KNN ($k$ = 10) | 1 | 101 | 81 | 100 | 37 | 117 | 61 | 145 | 46 | 76 | 60 | 36 | 58 | 75 | 78 | 3 | 4 | 79 |
| SNNC ($k_{BS}$) | 104 | 3 | 1 | 37 | 77 | 58 | 38 | 2 | 61 | 119 | 60 | 5 | 6 | 100 | 82 | 76 | 40 | 36 |
| SNNC ($k$ = 3) | 180 | 49 | 84 | 65 | 132 | 10 | 6 | 81 | 77 | 142 | 21 | 2 | 1 | 88 | 82 | 207 | 57 | 76 |
| uniform BS | 1 | 14 | 27 | 40 | 53 | 66 | 79 | 92 | 105 | 118 | 131 | 144 | 157 | 170 | 183 | 196 | 209 | 220 |

In order to demonstrate the effectuveness of BDPC, HSIC was used to illustrate its application, where IEPF-G-g was implemented as the desired classifier. Figure 7a–r shows the classification maps produced by (a) full bands, (b) UBS, (c) $b_c$-BDPC-SID, (d) $b_c$-BDPC-SAM, (e) $b_c$-BDPC-SIDAM, (f) $k$-BDPC-SID, (g) $k$-BDPC-SAM, (h) $k$-BDPC-SIDAM, (i) SMI-BS, (j) ECA, (k) E-FDPC, (l) IaDPI, (m) DPC-KNN($k_{BS}$), (n) DPC-KNN ($k$ = 2%L), (o) G-DPC-KNN ($k_{BS}$), (p) G-DPC-KNN ($k$ = 10), (q) SNNC($k_{BS}$), and (r) SNNC ($k$ = 3), respectively. Upon visual inspection, all the methods seemed to perform reasonably well on classification.

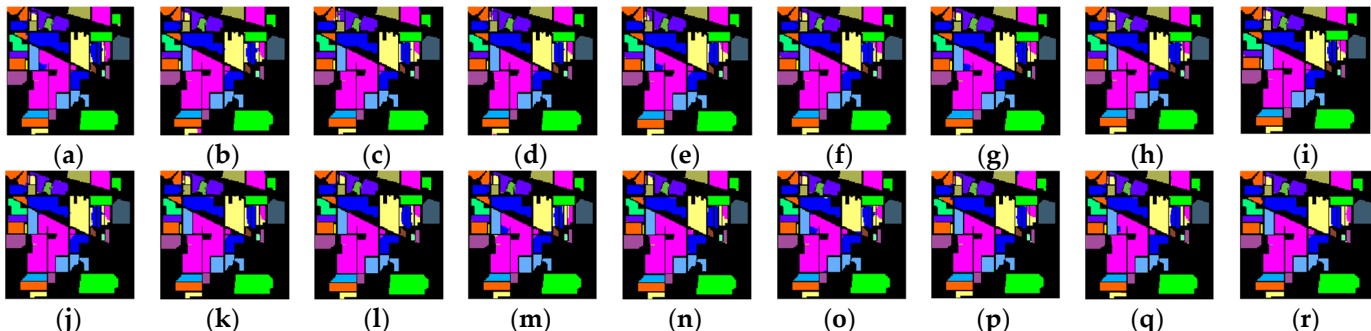

**Figure 7.** IEPF-G-g classification maps produced by different BS methods for Purdue data. (**a**) Full bands (OA = 95.71%), (**b**) UBS (OA = 96.81%), (**c**) $b_c$-BDPC-SID (OA = 98.08%),(**d**) $b_c$-BDPC-SAM (OA = 98.12%), (**e**) $b_c$-BDPC-SIDAM (OA =97.70%), (**f**) $k$-BDPC-SID (OA = 97.53%), (**g**) $k$ -BDPC-SAM (OA = 97.91%), (**h**) $k$-BDPC-SIDA (OA = 97.78%), (**i**) SMI-BS (OA = 97.57%), (**j**) ECA (OA = 96.87%), (**k**) E-FDPC (OA = 97.80%), (**l**) IaDPI (OA = 97.59%), (**m**) DPC-KNN ($k_{BS}$) (OA = 91.77%), (**n**) DPC-KNN ($k$ = 2%$L$) (OA = 98.05%), (**o**) G-DPC-KNN ($k_{BS}$) (OA = 89.33%), (**p**) G-DPC-KNN ($k$ = 10) (OA = 97.65%), (**q**) SNNC ($k_{BS}$) (OA = 97.14%), (**r**) SNNC($k$ = 3) (OA = 97.66%).

However, there are some subtle and appreciable differences in the classification of class 2 (highlighted in blue), class 10 (highlighted in yellow), and class 11 (highlighted in purple), where different methods misclassified different data samples. So, assessing which one performs better on these classes is a challenge. This is because the classification maps in Figure 7 only provide qualitative analyses, and it is still very difficult to evaluate the overall effectiveness of different methods based only on their classification maps.

To address this issue, Table 7 tabulates the classification results of $b_c$–BDPC and $k$-BDPC using SID, SAM, and SIDAM, along with the results produced by full bands and uniform BS for quantitative comparison. The commonly used overall accuracy (OA), average accuracy (AA), and Kappa coefficient were used as evaluation criteria, with the best results boldfaced in red and the second best boldfaced in black. As we can see, the best result was produced

by $b_c$-BDPC using SID, which also outperformed using full bands and uniform BS. Most impressively, all the various versions of $b_c$-BDPC and $k$-BDPC performed better than full bands and uniform BS, albeit at the expense of additional computer running time (seconds), documented at the bottom row of Table 7 for reference.

**Table 7.** Quantitative comparative analysis of classification for the Purdue data among $b_c$-BDPC and $k$-BDPC using, SID, SAM, and SIDAM along with full bands and uniform BS, with the best and second-best results highlighted and boldfaced in red and black, respectively.

| | IEPF-G-g | | | | | | | |
|---|---|---|---|---|---|---|---|---|
| **Class** | $b_c$-**BDPC-SID** | $b_c$-**BDPC-SAM** | $b_c$-**BDPC-SIDAM** | $k$-**BDPC-SID** | $k$-**BDPC-SAM** | $k$-**BDPC-SIDAM** | **Uniform BS** | **Full Bands** |
| 1 | 1.0000 | 0.9957 | 1.0000 | 1.0000 | 1.0000 | 1.0000 | 0.9978 | 0.9957 |
| 2 | 0.9646 | 0.9660 | 0.9644 | 0.9590 | 0.9637 | 0.9568 | 0.9293 | 0.9302 |
| 3 | 0.9886 | 0.9833 | 0.9865 | 0.9857 | 0.9877 | 0.9883 | 0.9839 | 0.9619 |
| 4 | 0.9975 | 0.9987 | 0.9979 | 0.9983 | 0.9992 | 0.9987 | 1.0000 | 0.9899 |
| 5 | 0.9783 | 0.9799 | 0.9783 | 0.9762 | 0.9785 | 0.9756 | 0.9764 | 0.9627 |
| 6 | 0.9996 | 0.9993 | 0.9999 | 0.9996 | 0.9997 | 0.9993 | 0.9986 | 0.9947 |
| 7 | 0.9929 | 0.9893 | 0.9964 | 0.9929 | 0.9929 | 0.9929 | 0.9893 | 0.9893 |
| 8 | 1.0000 | 1.0000 | 1.0000 | 1.0000 | 1.0000 | 1.0000 | 1.0000 | 1.0000 |
| 9 | 1.0000 | 1.0000 | 1.0000 | 1.0000 | 1.0000 | 1.0000 | 1.0000 | 1.0000 |
| 10 | 0.9697 | 0.9786 | 0.9643 | 0.9770 | 0.9773 | 0.9731 | 0.9551 | 0.9399 |
| 11 | 0.9656 | 0.9654 | 0.9527 | 0.9446 | 0.9559 | 0.9530 | 0.9411 | 0.9224 |
| 12 | 0.9921 | 0.9874 | 0.9892 | 0.9914 | 0.9917 | 0.9921 | 0.9916 | 0.9855 |
| 13 | 0.9976 | 0.9961 | 0.9971 | 0.9976 | 0.9971 | 0.9976 | 0.9966 | 0.9966 |
| 14 | 0.9962 | 0.9975 | 0.9972 | 0.9963 | 0.9972 | 0.9963 | 0.9962 | 0.9837 |
| 15 | 0.9969 | 0.9940 | 0.9982 | 0.9951 | 0.9953 | 0.9953 | 0.9969 | 0.9886 |
| 16 | 0.9968 | 0.9978 | 0.9978 | 0.9989 | 0.9978 | 0.9989 | 0.9989 | 0.9957 |
| $P_{OA}$ | **0.9808** | **0.9812** | 0.9770 | 0.9753 | 0.9791 | 0.9768 | 0.9681 | 0.9571 |
| $P_{AA}$ | **0.9898** | 0.9893 | 0.9887 | 0.9883 | 0.9896 | **0.9886** | 0.9845 | 0.9773 |
| Kappa | **0.9779** | **0.9786** | 0.9740 | 0.9721 | 0.9763 | 0.9736 | 0.9638 | 0.9514 |
| time (s) | 27 | 3 | 85 | 42 | 7 | 69 | | |

To further conduct a comparative analysis, Table 8 also tabulates the (OA, AA, Kappa) classification results obtained by the existing DPC-based BS methods: ECA, E-FDPC, IaDPI, DPC-KNN, G-DPC-KNN, and SNNC, where the best and second-best results are boldfaced in red and black, respectively, with running time (seconds) included at the bottom row for reference. Since SMI-BS has been shown to perform better than many existing BS methods in [5], SMI-BS results are also included in Table 8 for comparison. As shown in Table 8, E-FDPC and DPC-KNN using $k = 2\%L$ were the best. Nevertheless, SMI-BS, IaDPI, and SNNC also performed very well, with nearly the same performance as E-FDPC and DPC-KNN. Now, if we compare Table 7 to Table 8, we can see that $b_c$-BDPC using SID performed better than the best results of E-FDPC and DPC-KNN in Table 7.

### 3.2.2. Salinas

The VD estimated for Salinas was 21 [15,16]. Table 9 also includes 21 bands selected by SMI-BS, ECA, E-FDPC, IaDPI, DPC-kNN, G-DPC-kNN, SNNC, and uniform BS for comparison, where parameters used by various methods are specified in Table 5. Figures 8 and 9 plot $\eta_i^{b_c-\text{BDPC}}$ and $\eta_i^{k-\text{BDPC}}$ produced by $b_c$-BDPC and $k$-BDPC using (a) SID, (b) SAM, and (c) SIDAM for Salinas, respectively, where the red dots are the selected bands and are tabulated in Table 9.

**Table 8.** Quantitative comparative analysis of classification for the Purdue data among SMI-BS, ECA, E-FDPC, IaDPI, DPC-KNN, G-DPC-KNN, and SNNC, with the best and second-best results highlighted and boldfaced in red and black, respectively.

| | | | | | IEPF-G-g | | | | | |
|---|---|---|---|---|---|---|---|---|---|---|
| Class | SMI-BS | ECA | E-FDPC | IaDPI | DPC-KNN ($k_{BS}$) | DPC-KNN ($k = 2\%L$) | G-DPC-KNN ($k_{BS}$) | G-DPC-KNN ($k = 10$) | SNNC ($k_{BS}$) | SNNC ($k = 3$) |
| 1 | 1.0000 | 1.0000 | 0.9978 | 0.9978 | 1.0000 | 0.9957 | 0.9609 | 0.9978 | 0.9957 | 0.9957 |
| 2 | 0.9575 | 0.9440 | 0.9629 | 0.9538 | 0.8965 | 0.9723 | 0.8763 | 0.9567 | 0.9502 | 0.9491 |
| 3 | 0.9851 | 0.9800 | 0.9841 | 0.9878 | 0.9167 | 0.9898 | 0.8848 | 0.9837 | 0.9855 | 0.9887 |
| 4 | 0.9987 | 0.9987 | 0.9975 | 0.9975 | 0.9764 | 0.9975 | 0.9435 | 0.9992 | 0.9970 | 0.9970 |
| 5 | 0.9797 | 0.9783 | 0.9762 | 0.9816 | 0.9482 | 0.9723 | 0.8874 | 0.9716 | 0.9712 | 0.9789 |
| 6 | 0.9993 | 0.9981 | 0.9984 | 0.9989 | 0.9296 | 0.9970 | 0.8788 | 0.9964 | 0.9968 | 0.9989 |
| 7 | 0.9929 | 0.9893 | 0.9964 | 0.9929 | 1.0000 | 0.9857 | 0.9750 | 0.9857 | 0.9857 | 0.9964 |
| 8 | 1.0000 | 1.0000 | 1.0000 | 1.0000 | 0.9862 | 0.9998 | 0.9427 | 1.0000 | 1.0000 | 1.0000 |
| 9 | 1.0000 | 1.0000 | 1.0000 | 1.0000 | 0.9650 | 1.0000 | 0.9350 | 1.0000 | 1.0000 | 1.0000 |
| 10 | 0.9679 | 0.9531 | 0.9779 | 0.9723 | 0.9032 | 0.9767 | 0.8606 | 0.9677 | 0.9629 | 0.9660 |
| 11 | 0.9505 | 0.9374 | 0.9545 | 0.9509 | 0.8727 | 0.9604 | 0.9124 | 0.9576 | 0.9431 | 0.9598 |
| 12 | 0.9922 | 0.9921 | 0.9921 | 0.9934 | 0.9290 | 0.9931 | 0.8663 | 0.9944 | 0.9933 | 0.9907 |
| 13 | 0.9951 | 0.9971 | 0.9966 | 0.9961 | 0.9815 | 0.9976 | 0.9444 | 0.9980 | 0.9980 | 0.9980 |
| 14 | 0.9963 | 0.9947 | 0.9968 | 0.9947 | 0.9435 | 0.9957 | 0.8736 | 0.9960 | 0.9942 | 0.9953 |
| 15 | 0.9953 | 0.9974 | 0.9917 | 0.9959 | 0.9733 | 0.9876 | 0.9254 | 0.9899 | 0.9878 | 0.9959 |
| 16 | 0.9989 | 1.0000 | 0.9968 | 0.9978 | 0.9645 | 0.9925 | 0.9570 | 0.9978 | 0.9957 | 0.9978 |
| $P_{OA}$ | 0.9757 | 0.9687 | **0.9780** | 0.9759 | 0.9177 | **0.9805** | 0.8933 | 0.9765 | 0.9714 | 0.9766 |
| $P_{AA}$ | 0.9881 | 0.9850 | **0.9887** | 0.9882 | 0.9491 | **0.9883** | 0.9140 | 0.9870 | 0.9848 | 0.9880 |
| Kappa | 0.9723 | 0.9644 | **0.9751** | 0.9724 | 0.9071 | **0.9779** | 0.8798 | 0.9734 | 0.9677 | 0.9734 |
| time (s) | 36 | 0.2 | 0.2 | 0.3 | 0.2 | 0.2 | 0.3 | 0.3 | 0.4 | 0.7 |

**Table 9.** 21 bands selected for Salinas.

| | Selected Bands ($n_{VD} = 21$) | | | | | | | | | | | | | | | | | | | | |
|---|---|---|---|---|---|---|---|---|---|---|---|---|---|---|---|---|---|---|---|---|---|
| $b_c$-BDPC-SID | 91 | 141 | 157 | 156 | 155 | 70 | 159 | 108 | 192 | 164 | 160 | 165 | 31 | 111 | 166 | 110 | 163 | 154 | 224 | 18 | 52 |
| $b_c$-BDPC -SAM | 92 | 135 | 156 | 70 | 162 | 187 | 161 | 109 | 165 | 110 | 111 | 166 | 154 | 31 | 112 | 52 | 18 | 108 | 159 | 107 | 153 |
| $b_c$-BDPC -SIDAM | 92 | 156 | 155 | 157 | 164 | 159 | 163 | 160 | 161 | 165 | 108 | 135 | 111 | 154 | 166 | 192 | 70 | 110 | 112 | 224 | 31 |
| $k$-BDPC-SID | 157 | 160 | 159 | 164 | 155 | 108 | 163 | 165 | 110 | 162 | 111 | 154 | 166 | 224 | 1 | 112 | 222 | 107 | 167 | 113 | 2 |
| $k$-BDPC -SAM | 156 | 164 | 108 | 160 | 155 | 154 | 109 | 165 | 159 | 110 | 166 | 111 | 157 | 163 | 112 | 224 | 153 | 162 | 167 | 107 | 1 |
| $k$-BDPC -SIDAM | 156 | 164 | 159 | 160 | 155 | 157 | 108 | 163 | 165 | 162 | 110 | 111 | 154 | 166 | 224 | 112 | 222 | 107 | 167 | 1 | 113 |
| SMI-BS | 15 | 32 | 34 | 40 | 63 | 117 | 150 | 157 | 184 | 206 | 223 | 22 | 41 | 64 | 71 | 94 | 118 | 151 | 176 | 185 | 202 |
| ECA | 109 | 184 | 88 | 68 | 136 | 211 | 32 | 194 | 218 | 55 | 215 | 96 | 153 | 93 | 118 | 204 | 224 | 160 | 161 | 166 | 180 |
| E-FDPC | 193 | 61 | 104 | 5 | 4 | 103 | 78 | 64 | 81 | 102 | 9 | 39 | 105 | 2 | 36 | 83 | 38 | 3 | 80 | 106 | 84 |
| IaDPI | 154 | 38 | 107 | 194 | 8 | 65 | 140 | 19 | 89 | 167 | 101 | 112 | 159 | 136 | 96 | 33 | 85 | 133 | 175 | 170 | 162 |
| DPC-KNN ($k_{BS}$) | 111 | 24 | 89 | 73 | 197 | 129 | 104 | 5 | 17 | 105 | 64 | 2 | 63 | 210 | 81 | 3 | 37 | 15 | 36 | 106 | 103 |
| DPC-KNN ($k = 6\%L$) | 159 | 156 | 157 | 158 | 163 | 162 | 164 | 108 | 165 | 110 | 111 | 155 | 154 | 112 | 222 | 212 | 137 | 220 | 92 | 161 | 217 |
| G-DPC-KNN ($k_{BS}$) | 204 | 205 | 202 | 150 | 209 | 206 | 207 | 119 | 203 | 118 | 117 | 170 | 36 | 22 | 23 | 24 | 27 | 21 | 17 | 15 | 25 |
| G-DPC-KNN ($k = 7$) | 38 | 64 | 57 | 49 | 6 | 134 | 133 | 63 | 108 | 109 | 5 | 104 | 45 | 43 | 78 | 8 | 87 | 38 | 64 | 57 | 49 |
| SNNC ($k_{BS}$) | 39 | 121 | 4 | 22 | 3 | 117 | 123 | 41 | 104 | 63 | 85 | 43 | 64 | 1 | 2 | 65 | 8 | 39 | 121 | 4 | 22 |
| SNNC ($k = 3$) | 183 | 5 | 130 | 14 | 18 | 173 | 24 | 4 | 84 | 3 | 66 | 96 | 82 | 193 | 10 | 2 | 54 | 183 | 5 | 130 | 14 |
| Uniform BS | 1 | 12 | 23 | 34 | 45 | 56 | 67 | 78 | 89 | 100 | 111 | 122 | 133 | 144 | 155 | 166 | 177 | 188 | 199 | 210 | 224 |

Figure 10a–r shows the classification maps produced by (a) full bands, (b) UBS, (c) $b_c$-BDPC-SID, (d) $b_c$-BDPC-SAM, (e) $b_c$-BDPC-SIDAM, (f) $k$-BDPC-SID, (g) $k$-BDPC-SAM, (h) $k$-BDPC-SIDAM, (i) SMI-BS, (j) ECA, (k) E-FDPC, (l) IaDPI, (m) DPC-KNN, ($k_{BS}$) (n) DPC-KNN ($k = 2\%L$), (o) G-DPC-KNN ($k_{BS}$), (p) G-DPC-KNN ($k = 10$), (q) SNNC ($k_{BS}$), and (r) SNNC ($k = 3$), respectively. Like the Purdue data, all the methods seemed to perform comparably on classification except for the two largest classes: class 8 (highlighted in dark blue) and class 15 (highlighted in brown), where different methods showed visible differences. In particular, (c) $b_c$-BDPC-SID, (p) G-DPC-KNN ($k = 10$), and (r) SNNC ($k = 3$) were among the best performers for the classification of classes 8 and 15. However, this was not true when the overall performance was considered.

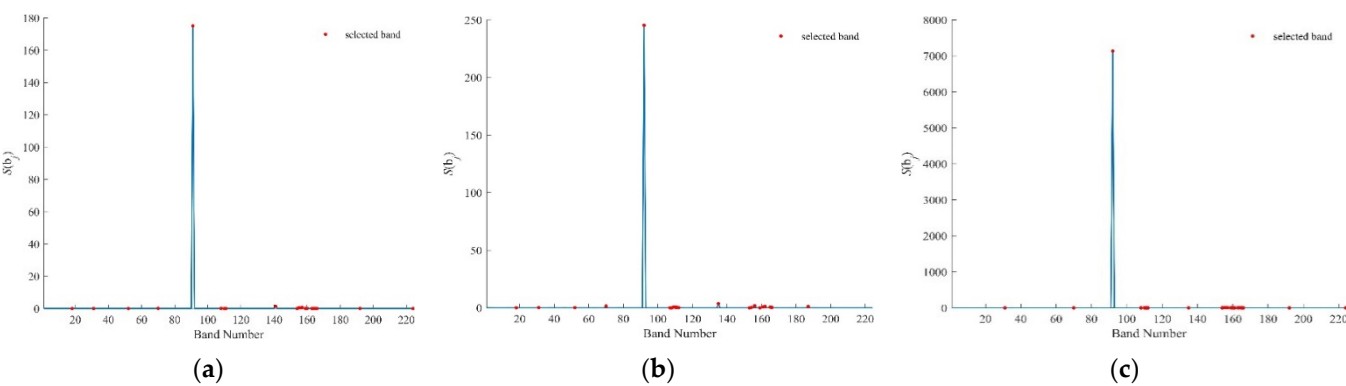

**Figure 8.** $\eta_i^{b_c-\text{BDPC}}$ values by $b_c$-BDPC for Salinas using (**a**) SID, (**b**) SAM, (**c**) SIDAM.

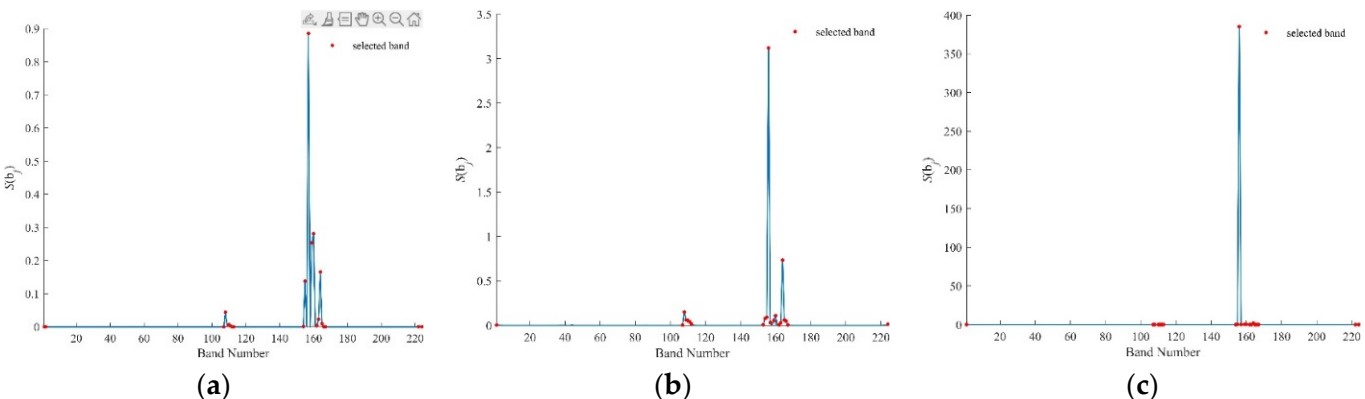

**Figure 9.** $\eta_i^{k-\text{BDPC}}$ values by $k$-BDPC for Salinas using (**a**) SID, (**b**) SAM, (**c**) SIDAM.

Table 10 tabulates the (OA, AA, Kappa) classification results of $b_c$-BDPC and $k$-BDPC using, SID, SAM, and SIDAM, along with the results produced by full bands and uniform BS for quantitative comparison, where the best and second-best results are boldfaced in red and black, respectively. As shown in Table 10, the best results were produced by $b_c$-BDPC using SAM and full bands. Interestingly, unlike the Purdue data, all the various versions of $b_c$-BDPC and $k$-BDPC did not perform as well as full bands and uniform BS. The quantitative results in Table 10 demonstrated that the qualitative visual inspection in Figure 10 was not reliable.

To further conduct a comparative analysis, Table 11 also tabulates the (OA, AA, Kappa) classification results obtained by ECA, E-FDPC, IaDPI, DPC-KNN, G-DPC-KNN, and SNNC, where the best and second-best results are boldfaced in red and black, respectively, with running time (seconds) included at the bottom row for reference. Since SMI-BS has been shown to perform better than many existing BS methods in [5], SMI-BS results are also included in Table 11 for comparison.

As shown in Table 11, SNNC ($k$ = 3) was the best, followed by DPC-KNN using $k_{\text{BS}}$ as the second-best. Interestingly, SNNC ($k$ = 3) not only outperformed all the DPC-based methods in Table 11 but also performed better than using full bands. According to Tables 10 and 11, it is worth noting that all the methods had difficulty with classifying classes 8 and 15, which are adjacent each other. This is particularly true for $k$-BDPC. Because of the poor classification of these two largest classes, $k$-BDPC did not perform well compared to other methods.

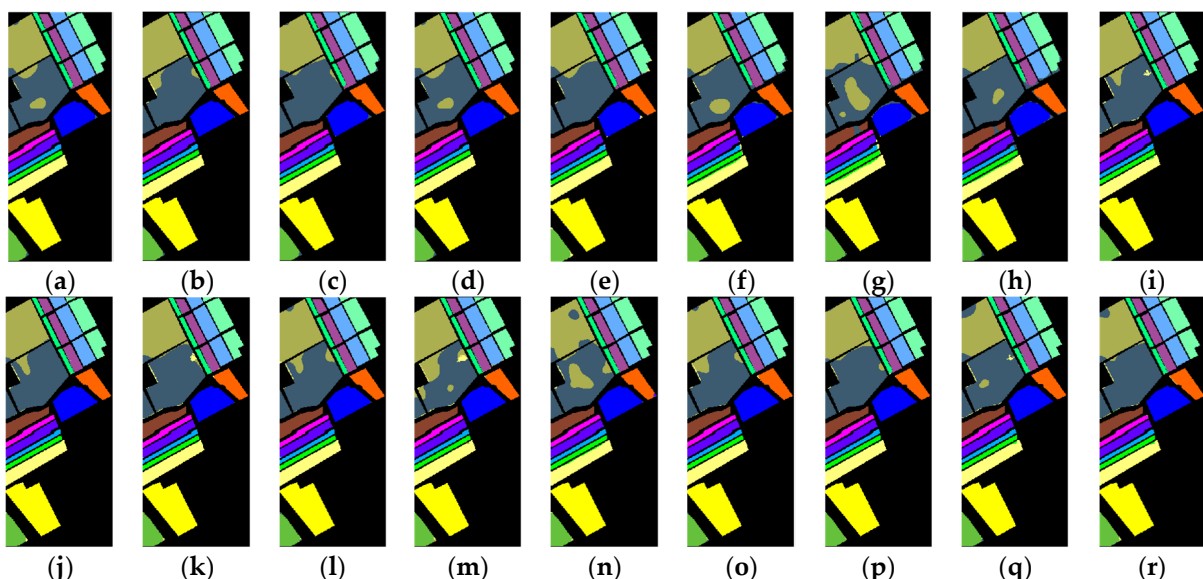

**Figure 10.** Classification maps produced by IEPF-G-g using different BS criterion for Salinas data. (**a**) Full bands (OA = 96.24%), (**b**) UBS (OA = 95.69%), (**c**) $b_c$-BDPC-SID (OA = 95.28%), (**d**) $b_c$-BDPC-SAM (OA = 96.32%), (**e**) $b_c$-BDPC-SIDAM (OA = 96.09%), (**f**) $k$-BDPC-SID (OA = 91.89%), (**g**) $k$ -BDPC-SAM (OA = 90.09%), (**h**) $k$-BDPC-SIDAM (OA = 90.50%), (**i**) SMI-BS (OA = 95.32%), (**j**) ECA (OA = 96.32%), (**k**) E-FDPC (OA =96.47%), (**l**) IaDPI (OA = 95.86%), (**m**) DPC-KNN ($k_{BS}$) (OA =96.86%), (**n**) DPC-KNN ($k$ = 2%$L$) (OA = 95.16%), (**o**) G-DPC-KNN ($k_{BS}$) (OA = 95.05%), (**p**) G-DPC-KNN ($k$ = 10) (OA = 95.81%), (**q**) SNNC ($k_{BS}$) (OA = 95.29%), (**r**) SNNC ($k$ = 3) (OA = 96.91%).

**Table 10.** Quantitative comparative analysis of classification for Salinas among $b_c$-BDPC and $k$-BDPC using SID, SAM, and SIDAM, along with full bands and uniform BS, with the best and second-best results highlighted and boldfaced in red and black, respectively.

| | IEPF-G-g | | | | | | | |
|---|---|---|---|---|---|---|---|---|
| Class | $b_c$-BDPC-SID | $b_c$-BDPC-SAM | $b_c$-BDPC-SIDAM | $k$-BDPC-SID | $k$-BDPC-SAM | $k$-BDPC-SIDAM | UBS | Full Bands |
| 1 | 0.9990 | 0.9979 | 0.9983 | 0.9924 | 0.9855 | 0.9795 | 0.9998 | 1.0000 |
| 2 | 0.9909 | 0.9914 | 0.9937 | 0.9837 | 0.9749 | 0.9612 | 0.9964 | 0.9967 |
| 3 | 0.9989 | 0.9978 | 0.9935 | 0.9945 | 0.9907 | 0.9844 | 0.9997 | 0.9986 |
| 4 | 0.9945 | 0.9944 | 0.9963 | 0.9919 | 0.9896 | 0.9851 | 0.9986 | 0.9973 |
| 5 | 0.9872 | 0.9886 | 0.9851 | 0.9757 | 0.9801 | 0.9786 | 0.9916 | 0.9899 |
| 6 | 0.9961 | 0.9962 | 0.9946 | 0.9983 | 0.9997 | 0.9937 | 0.9988 | 0.9978 |
| 7 | 0.9943 | 0.9926 | 0.9920 | 0.9940 | 0.9944 | 0.9806 | 0.9963 | 0.9958 |
| 8 | 0.8876 | 0.9058 | 0.9048 | 0.7649 | 0.7155 | 0.7559 | 0.8886 | 0.8944 |
| 9 | 0.9946 | 0.9929 | 0.9848 | 0.9934 | 0.9903 | 0.9893 | 0.9957 | 0.9964 |
| 10 | 0.9868 | 0.9883 | 0.9739 | 0.9738 | 0.9702 | 0.9612 | 0.9889 | 0.9820 |
| 11 | 0.9934 | 0.9925 | 0.9944 | 0.9949 | 0.9966 | 0.9829 | 0.9982 | 0.9960 |
| 12 | 0.9992 | 0.9977 | 0.9982 | 0.9866 | 0.9955 | 0.9853 | 1.0000 | 1.0000 |
| 13 | 0.9893 | 0.9902 | 0.9880 | 0.9988 | 0.9979 | 0.9848 | 0.9927 | 0.9936 |
| 14 | 0.9874 | 0.9904 | 0.9880 | 0.9931 | 0.9857 | 0.9859 | 0.9932 | 0.9894 |
| 15 | 0.8968 | 0.9084 | 0.9098 | 0.8127 | 0.7621 | 0.7646 | 0.8728 | 0.9098 |
| 16 | 0.9878 | 0.9826 | 0.9776 | 0.9872 | 0.9925 | 0.9877 | 0.9903 | 0.9901 |
| $P_{OA}$ | 0.9582 | **0.9632** | 0.9609 | 0.9189 | 0.9009 | 0.9050 | 0.9569 | **0.9624** |
| $P_{AA}$ | 0.9802 | **0.9817** | 0.9796 | 0.9647 | 0.9576 | 0.9538 | 0.9814 | **0.9830** |
| Kappa | 0.9537 | **0.9529** | 0.9567 | 0.9106 | 0.8911 | 0.8955 | 0.9523 | **0.9583** |
| time (s) | 139 | 8 | 275 | 228 | 48 | 278 | | |

**Table 11.** Quantitative comparative analysis of classification for Salinas among SMI-BS, ECA, E-FDPC, IaDPI, DPC-KNN, G-DPC-KNN, and SNNC, with the best and second-best results highlighted and boldfaced in red and black, respectively.

| Class | SMI-BS | ECA | E-FDPC | IaDPI | DPC-KNN ($k_{BS}$) | DPC-KNN ($k = 2\%L$) | G-DPC-KNN ($k_{BS}$) | G-DPC-KNN ($k = 7$) | SNNC ($k_{BS}$) | SNN ($k = 3$) |
|---|---|---|---|---|---|---|---|---|---|---|
| | | | | | | **IEPF-G-g** | | | | |
| 1 | 0.9998 | 0.9999 | 1.0000 | 1.0000 | 0.9999 | 0.9959 | 0.9940 | 0.9999 | 1.0000 | 0.9999 |
| 2 | 0.9947 | 0.9948 | 0.9979 | 0.9989 | 0.9977 | 0.9857 | 0.9977 | 0.9973 | 0.9979 | 0.9976 |
| 3 | 0.9996 | 0.9991 | 0.9992 | 0.9992 | 0.9999 | 0.9920 | 0.9879 | 0.9990 | 0.9997 | 0.9997 |
| 4 | 0.9979 | 0.9981 | 0.9966 | 0.9985 | 0.9958 | 0.9946 | 0.9951 | 0.9981 | 0.9976 | 0.9961 |
| 5 | 0.9909 | 0.9874 | 0.9896 | 0.9879 | 0.9929 | 0.9825 | 0.9771 | 0.9884 | 0.9882 | 0.9926 |
| 6 | 0.9982 | 0.9978 | 0.9994 | 0.9987 | 0.9990 | 0.9930 | 0.9922 | 0.9990 | 0.9993 | 0.9987 |
| 7 | 0.9955 | 0.9960 | 0.9968 | 0.9963 | 0.9963 | 0.9907 | 0.9928 | 0.9956 | 0.9961 | 0.9963 |
| 8 | 0.8611 | 0.9004 | 0.8997 | 0.8773 | 0.9178 | 0.8784 | 0.9037 | 0.8710 | 0.8652 | 0.9160 |
| 9 | 0.9945 | 0.9950 | 0.9998 | 0.9963 | 0.9960 | 0.9867 | 0.9880 | 0.9977 | 0.9993 | 0.9968 |
| 10 | 0.9822 | 0.9824 | 0.9829 | 0.9884 | 0.9854 | 0.9678 | 0.9803 | 0.9855 | 0.9864 | 0.9882 |
| 11 | 0.9961 | 0.9947 | 0.9963 | 0.9970 | 0.9978 | 0.9898 | 0.9890 | 0.9965 | 0.9971 | 0.9970 |
| 12 | 1.0000 | 0.9996 | 1.0000 | 0.9999 | 1.0000 | 0.9884 | 0.9934 | 1.0000 | 1.0000 | 0.9999 |
| 13 | 0.9936 | 0.9940 | 0.9963 | 0.9976 | 0.9931 | 0.9922 | 0.9956 | 0.9944 | 0.9951 | 0.9942 |
| 14 | 0.9915 | 0.9920 | 0.9951 | 0.9933 | 0.9918 | 0.9841 | 0.9873 | 0.9928 | 0.9917 | 0.9931 |
| 15 | 0.8946 | 0.9091 | 0.9121 | 0.9026 | 0.9154 | 0.8946 | 0.8393 | 0.9095 | 0.8763 | 0.9201 |
| 16 | 0.9912 | 0.9883 | 0.9918 | 0.9888 | 0.9907 | 0.9736 | 0.9869 | 0.9939 | 0.9951 | 0.9900 |
| $P_{OA}$ | 0.9532 | 0.9632 | 0.9647 | 0.9586 | **0.9686** | 0.9516 | 0.9515 | 0.9581 | 0.9529 | **0.9691** |
| $P_{AA}$ | 0.9801 | 0.9830 | 0.9846 | 0.9826 | **0.9856** | 0.9744 | 0.9750 | 0.9824 | 0.9803 | **0.9860** |
| Kappa | 0.9483 | 0.9591 | 0.9609 | 0.9541 | **0.9652** | 0.9464 | 0.9462 | 0.9536 | 0.9478 | **0.9657** |
| time (s) | 149 | 1.8 | 1.8 | 2.1 | 1.9 | 1.8 | 1.7 | 1.7 | 1.9 | 1.9 |

### 3.2.3. U. of Pavia

Once again, the VD estimated for U. of Pavia was 14 [15,16]. Figures 11 and 12 plot $\eta_i^{b_c-\text{BDPC}}$ and $\eta_i^{k-\text{BDPC}}$ produced by $b_c$-BDPC and $k$-BDPC, using (a) SID, (b) SAM, and (c) SIDAM for the U. of Pavia scene, respectively, where the red dots are the selected bands and are tabulated in Table 12. Table 12 also includes 14 bands selected by SMI-BS, ECA, E-FDPC, IaDPI, DPC-kNN, G-DPC-kNN, SNNC, and uniform BS.

**Table 12.** 14 bands selected for U. of Pavia.

| Selected Bands ($n_{VD} = 14$) | | | | | | | | | | | | | | |
|---|---|---|---|---|---|---|---|---|---|---|---|---|---|
| $b_c$-BDPC-SID | 93 | 62 | 1 | 2 | 3 | 71 | 4 | 70 | 32 | 5 | 72 | 69 | 6 | 7 |
| $b_c$-BDP-SAM | 91 | 55 | 31 | 19 | 84 | 1 | 2 | 82 | 58 | 50 | 72 | 59 | 3 | 81 |
| $b_c$-BDPC-SIDAM | 91 | 61 | 1 | 2 | 3 | 71 | 70 | 4 | 32 | 72 | 5 | 69 | 19 | 6 |
| $k$-BDPC-SID | 1 | 72 | 2 | 3 | 4 | 5 | 30 | 6 | 15 | 71 | 70 | 7 | 73 | 69 |
| $k$-BDPC-SAM | 1 | 72 | 2 | 3 | 4 | 27 | 5 | 71 | 73 | 70 | 74 | 6 | 69 | 16 |
| $k$-BDPC-SIDAM | 1 | 72 | 2 | 3 | 4 | 5 | 30 | 16 | 6 | 71 | 70 | 73 | 69 | 7 |
| SMI-BS | 22 | 41 | 91 | 9 | 18 | 21 | 37 | 40 | 48 | 57 | 66 | 82 | 92 | 94 |
| ECA | 61 | 88 | 53 | 46 | 33 | 62 | 56 | 63 | 49 | 60 | 92 | 47 | 57 | 54 |
| E-FDPC | 50 | 93 | 24 | 84 | 82 | 2 | 4 | 3 | 68 | 5 | 67 | 69 | 1 | 6 |
| IaDPI | 74 | 32 | 5 | 98 | 84 | 45 | 51 | 41 | 94 | 76 | 75 | 73 | 77 | 72 |
| DPC-KNN ($k_{BS}$) | 58 | 93 | 16 | 35 | 1 | 2 | 3 | 4 | 82 | 69 | 84 | 77 | 5 | 52 |
| DPC-KNN ($k = 2\%L$) | 60 | 63 | 61 | 57 | 53 | 47 | 64 | 51 | 49 | 45 | 89 | 32 | 91 | 55 |
| G-DPC-KNN ($k_{BS}$) | 28 | 27 | 29 | 26 | 30 | 31 | 36 | 35 | 33 | 34 | 32 | 79 | 78 | 77 |
| G-DPC-KNN ($k = 7$) | 75 | 71 | 72 | 68 | 67 | 76 | 74 | 69 | 73 | 60 | 58 | 56 | 50 | 53 |
| SNNC ($k_{BS}$) | 82 | 5 | 2 | 49 | 1 | 3 | 4 | 7 | 29 | 9 | 6 | 93 | 11 | 86 |
| SNNC ($k = 5$) | 92 | 59 | 1 | 7 | 2 | 3 | 4 | 17 | 5 | 100 | 6 | 75 | 8 | 9 |
| Uniform BS | 1 | 9 | 17 | 25 | 33 | 41 | 49 | 57 | 65 | 73 | 81 | 89 | 97 | 103 |

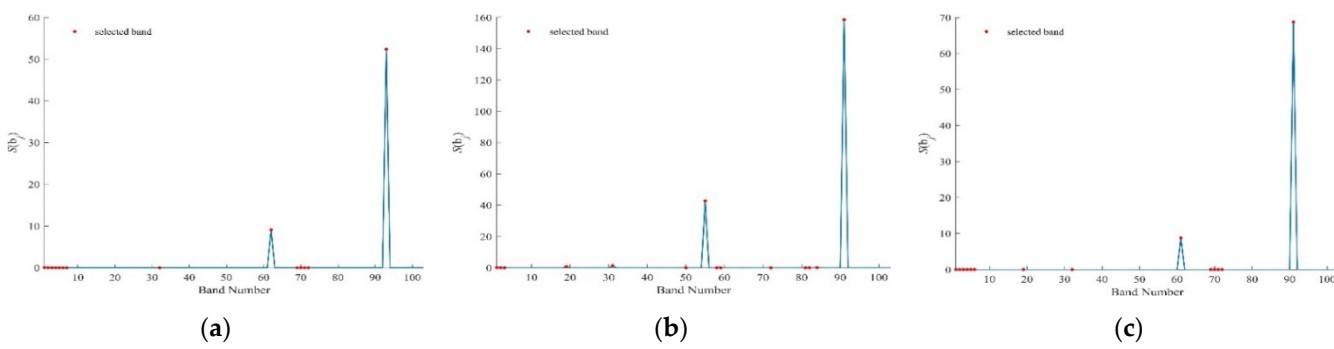

**Figure 11.** $\eta_i^{b_c-\text{BDPC}}$ values by $b_c$-BDPC for U. of Pavia using (**a**) SID, (**b**) SAM, (**c**) SIDAM.

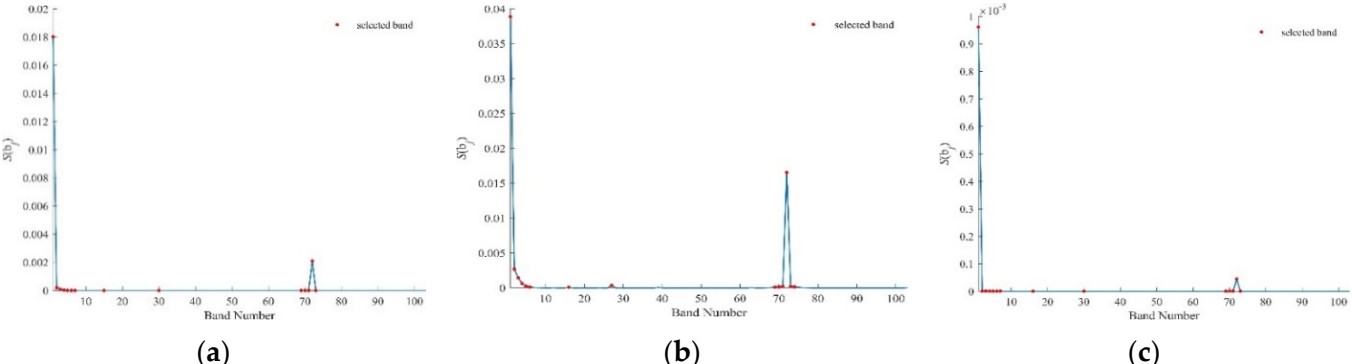

**Figure 12.** $\eta_i^{k-\text{BDPC}}$ values by $k$-BDPC for U. of Pavia using (**a**) SID, (**b**) SAM, (**c**) SIDAM.

Figure 13a–r shows the classification maps produced by (a) full bands, (b) UBS, (c) $b_c$-BDPC-SID, (d) $b_c$-BDPC-SAM, (e) $b_c$-BDPC-SIDAM, (f) $k$-BDPC-SID, (g) $k$-BDPC-SAM, (h) $k$-BDPC-SIDAM, (i) SMI-BS, (j) ECA, (k) E-FDPC, (l) IaDPI, (m) DPC-KNN($k_{\text{BS}}$), (n) DPC-KNN ($k$ = 2%L), (o) G-DPC-KNN ($k_{\text{BS}}$), (p) G-DPC-KNN ($k$ = 10), (q) SNNC ($k_{\text{BS}}$), and (r) SNNC ($k$ = 3), respectively. Overall, all the methods performed reasonably well upon visual inspection, except for the classification of the second largest class: class 6 (highlighted in brown), which had subtle differences.

However, to evaluate overall performance, it is extremely difficult to name which method is the best. In this case, Table 13 tabulates the (OA, AA, Kappa) classification results of $b_c$-BDPC and $k$-BDPC using SID, SAM, and SIDAM, along with the results produced by full bands and uniform BS for quantitative comparison with the best results boldfaced in red and the second-best results boldfaced in black. As we can see, the best results were produced using full bands, followed very closely by $b_c$-BDPC using SIDAM with only differences within 0.2%. Table 14 also tabulates the (OA, AA, Kappa) classification results obtained by ECA, E-FDPC, IaDPI, DPC-KNN, G-DPC-KNN, and SNNC, where the best and second-best results are boldfaced in red and black, respectively, with computer running time in seconds included at the bottom row for reference. As shown in Table 14, E-FDPC was the best, followed by the second best DPC-KNN using $k_{\text{BS}}$. If we compare Table 13 to Table 14, $b_c$-BDPC using SAM and SIDAM performed slightly better in OA than E-FDPC but slightly worse in AA than E-FDPC. As a matter of fact, $b_c$-BDPC generally outperformed all other BS methods in Table 14.

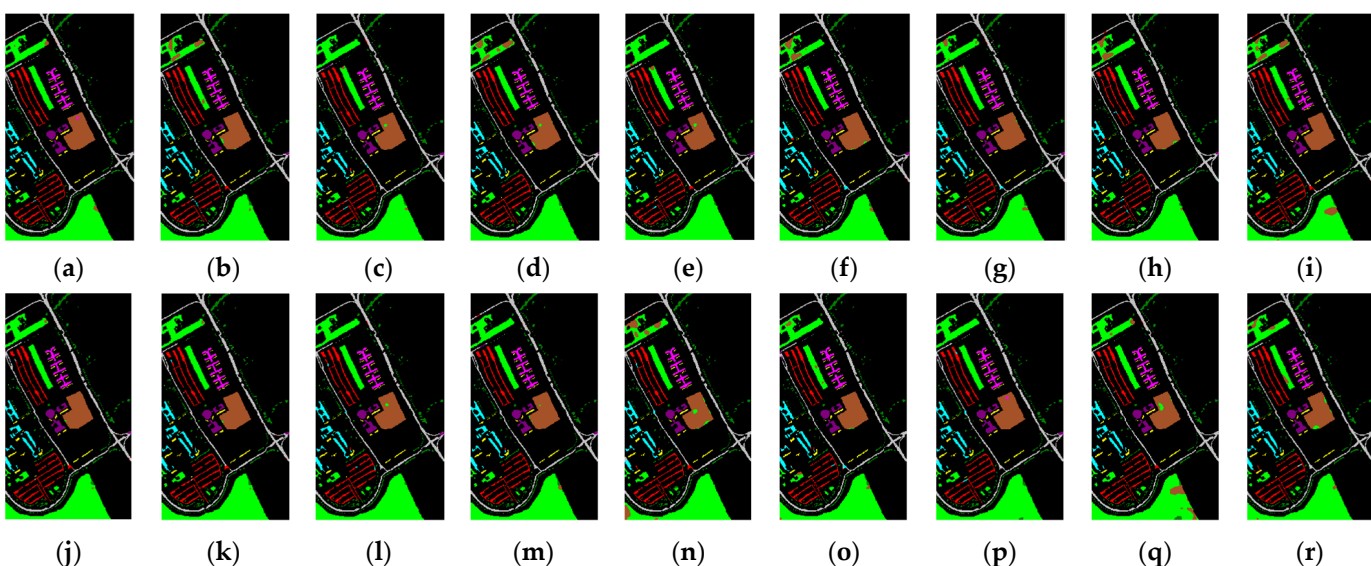

**Figure 13.** Classification maps produced by IEPF-G-g using different BS criterion for University of Pavia data. (**a**) Full bands (OA = 98.23%), (**b**) UBS (OA = 96.87%), (**c**) $b_c$-BDPC-SID (OA = 97.82%), (**d**) $b_c$-BDPC-SAM (OA = 97.83%), (**e**) $b_c$-BDPC-SIDAM (OA = 98.13%), (**f**) $k$-BDPC-SID (OA = 89.12%), (**g**) $k$-BDPC-SAM (OA = 9707%), (**h**) $k$-BDPC-SIDA (OA = 96.63%), (**i**) SMI-BS (OA = 97.46%), (**j**) ECA (OA = 95.76%), (**k**) E-FDPC (OA =97.95%), (**l**) IaDPI (OA = 97.00%), (**m**) DPC-KNN ($k_{BS}$) (OA = 97.79%), (**n**) DPC-KNN ($k$ = 2%$L$) (OA = 96.07%), (**o**) G-DPC-KNN ($k_{BS}$) (OA = 96.93%), (**p**) G-DPC-KNN ($k$ = 10) (OA = 96.79%), (**q**) SNNC ($k_{BS}$) (OA = 97.39%), (**r**) SNNC ($k$ = 3) (OA = 96.30%).

**Table 13.** Quantitative comparative analysis of classification for U. of Pavia among $b_c$-BDPC and $k$-BDPC using SID, SAM, and SIDAM, along with full bands and uniform BS, with the best and second-best results highlighted and boldfaced in red and black, respectively.

| | IEPF-G-g | | | | | | | |
|---|---|---|---|---|---|---|---|---|
| Class | $b_c$-**BDPC-SID** | $b_c$-**BDPC-SAM** | $b_c$-**BDPC-SIDAM** | $k$-**BDPC-SID** | $k$-**BDPC-SAM** | $k$-**BDPC-SIDAM** | **Uniform BS** | **Full Bands** |
| 1 | 0.9765 | 0.9811 | 0.9845 | 0.9064 | 0.9861 | 0.9877 | 0.9827 | 0.9816 |
| 2 | 0.9708 | 0.9658 | 0.9751 | 0.8574 | 0.9558 | 0.9452 | 0.9434 | 0.9738 |
| 3 | 0.9784 | 0.9881 | 0.9842 | 0.9041 | 0.9818 | 0.9817 | 0.9874 | 0.9915 |
| 4 | 0.9887 | 0.9874 | 0.9865 | 0.9079 | 0.9803 | 0.9805 | 0.9929 | 0.9870 |
| 5 | 0.9992 | 0.9995 | 0.9996 | 0.9320 | 0.9989 | 0.9988 | 0.9977 | 0.9984 |
| 6 | 0.9919 | 0.9940 | 0.9898 | 0.8931 | 0.9771 | 0.9727 | 0.9832 | 0.9967 |
| 7 | 0.9998 | 0.9996 | 0.9999 | 0.9265 | 1.0000 | 0.9999 | 0.9969 | 0.9997 |
| 8 | 0.9706 | 0.9812 | 0.9712 | 0.9747 | 0.9685 | 0.9734 | 0.9881 | 0.9816 |
| 9 | 0.9982 | 0.9995 | 0.9992 | 0.9235 | 0.9955 | 0.9962 | 0.9960 | 0.9998 |
| $P_{OA}$ | 0.9782 | 0.9783 | **0.9813** | 0.8912 | 0.9707 | 0.9663 | 0.9683 | **0.9823** |
| $P_{AA}$ | 0.9860 | **0.9885** | 0.9878 | 0.9140 | 0.9826 | 0.9818 | 0.9854 | **0.9900** |
| Kappa | 0.9713 | 0.9715 | **0.9754** | 0.8602 | 0.9616 | 0.9558 | 0.9584 | **0.9767** |
| time (s) | 85 | 14 | 118 | 165 | 38 | 189 | | |

**Table 14.** Quantitative comparative analysis of classification for U. of Pavia among SMI-BS, ECA, E-FDPC, IaDPI, DPC-KNN, G-DPC-KNN, and SNNC, with the best and second-best results highlighted and boldfaced in red and black, respectively.

| | | | | | IEPF-G-g | | | | | |
|---|---|---|---|---|---|---|---|---|---|---|
| **Class** | **SMI-BS** | **ECA** | **E-FDPC** | **IaDPI** | **DPC-KNN ($k_{BS}$)** | **DPC-KNN ($k = 2\%L$)** | **G-DPC-KNN ($k_{BS}$)** | **G-DPC-KNN ($k = 7$)** | **SNNC ($k_{BS}$)** | **SNNC ($k = 5$)** |
| 1 | 0.9782 | 0.9701 | 0.9833 | 0.9777 | 0.9844 | 0.9766 | 0.9723 | 0.9703 | 0.9823 | 0.9783 |
| 2 | 0.9613 | 0.9360 | 0.9684 | 0.9508 | 0.9658 | 0.9353 | 0.9559 | 0.9559 | 0.9584 | 0.9403 |
| 3 | 0.9879 | 0.9791 | 0.9884 | 0.9834 | 0.9841 | 0.9860 | 0.9870 | 0.9731 | 0.9871 | 0.9809 |
| 4 | 0.9933 | 0.9901 | 0.9898 | 0.9895 | 0.9898 | 0.9891 | 0.9876 | 0.9883 | 0.9919 | 0.9875 |
| 5 | 0.9967 | 0.9991 | 0.9991 | 0.9984 | 0.9980 | 0.9983 | 0.9984 | 0.9987 | 0.9981 | 0.9949 |
| 6 | 0.9873 | 0.9568 | 0.9922 | 0.9851 | 0.9923 | 0.9687 | 0.9705 | 0.9792 | 0.9847 | 0.9692 |
| 7 | 0.9972 | 0.9937 | 0.9992 | 0.9979 | 0.9987 | 0.9958 | 0.9968 | 0.9981 | 0.9986 | 0.9950 |
| 8 | 0.9733 | 0.9688 | 0.9784 | 0.9807 | 0.9751 | 0.9765 | 0.9769 | 0.9600 | 0.9758 | 0.9802 |
| 9 | 0.9957 | 0.9939 | 0.9987 | 0.9988 | 0.9961 | 0.9931 | 0.9974 | 0.9946 | 0.9971 | 0.9937 |
| $P_{OA}$ | 0.9746 | 0.9576 | **0.9795** | 0.9700 | **0.9779** | 0.9607 | 0.9693 | 0.9679 | 0.9739 | 0.9630 |
| $P_{AA}$ | 0.9857 | 0.9764 | **0.9886** | 0.9847 | **0.9871** | 0.9799 | 0.9825 | 0.9798 | 0.9860 | 0.9800 |
| Kappa | 0.9666 | 0.9446 | **0.9730** | 0.9606 | **0.9709** | 0.9486 | 0.9598 | 0.9579 | 0.9656 | 0.9516 |
| time(s) | 92 | 1.1 | 1.1 | 1.8 | 1.0 | 1.1 | 1.0 | 1.1 | 1.4 | 1.4 |

### 3.2.4. Discussions

The experiments in this section yield several interesting observations and findings.

- Experimental results showed that the three datasets exhibited different characteristics, which resulted in different performances of various BS methods and classifiers. For example, the Purdue data had four small classes with less than 100 data samples and three large classes with more than 1000 data samples, compared to Salinas and U. of Pavia, both of which had classes with more than 900 data samples. So, the Purdue data ran into an imbalanced class issue. As a result, full bands are generally good for balanced classes and performed better than all the test BS methods for Salinas and U. of Pavia but performed worse than SMI-BS and most DPC-based BS methods for the Purdue data.

- Also, interestingly, both $b_c$-BDPC and $k$-BDPC performed very well and better than full bands for the Purdue data. This indicates that $b_c$-BDPC and $k$-BDPC may perform better for datasets with imbalanced classes. However, $b_c$-BDPC and $k$-BDPC did not perform as well as full bands for Salinas and U. of Pavia. Nevertheless, $b_c$-BDPC still managed to perform comparably to full bands. Surprisingly, $k$-BDPC did not perform well as expected, specifically for Salinas, as its performance significantly degraded due to the classification of classes 8 and 15.

- Since BDPC combines the advantages of the two indicators of DPC, cluster density ρ and cluster distance δ, and the band prominent peaks of SMI-BS, BPV, it performs better than DPC-based BS and SMI-BS, as expected.

- According to experiments, $b_c$-BDPC generally performed better than $k$-BDPC for all the three datasets. This means that the cut-off distance, $b_c$, is more crucial than the $k$ nearest neighbors in designing DPC-BS. More specifically, $b_c \to \rho_i^{b_c-\text{BLD}} \to \delta_i^{k-\text{BD}}$ is more effective than $k \to \delta_i^{k-\text{BD}} \to \rho_i^{b_c-\text{BLD}}$.

- DPC-KNN using $k_{BS}$ performed better than DPC-KNN using $k = 2\%L$ for Salinas and U. of Pavia but worse for the Purdue data. This demonstrates that DPC-KNN using $k_{BS}$ works well for large datasets compared to DPC-KNN using $k = 2\%L$, which performs better for small datasets. In addition, G-DPC-KNN using $k_{BS}$ performed better than DPC-KNN using $k = 2\%L$ for U. of Pavia but worse for the Purdue and Salinas data scenes.

- It is notewothy that experiments showed that DPC-KNN and G-DPC-KNN using $k_{BS}$ did not perform as well as DPC-KNN and G-DPC-KNN did for the Purdue data, but did perform better than DPC-KNN and G-DPC-KNN for U. of Pavia. This indicates that DPC-KNN and G-DPC-KNN may work well for datasets mixed with small and large classes, while DPC-KNN and G-DPC-KNN using $k_{BS}$ may work well for datasets with large classes. As for the Salinas data, they all performed comparably.

- Analogously to DPC-KNN and G-DPC-KNN with/without using $k_{BS}$ discussed above, SNNC using $k_{BS}$ performed better than SNNC using (48) for U. of Pavia but worse for the Purdue and Salinas data scenes. Nevertheless, both methods performed very closely.

- According to [54], the evaluation metrics, OA, AA, and Kappa coefficient used to evaluate classification performance are referred to as a priori measures or producer's accuracy, based on ground truth. There is another type of evaluation metric, called precision rate (PR), referred to as a posteriori measure or user's accuracy, based on classified data samples and addresses classification with background [55,56]. Since PR is generally used to evaluate object detection coupled with recall rate in computer vision, its usage and discussions are beyond the scope of this paper. So, no discussions of PR are included.

- As shown by experiments, our proposed $k$-BDPC may not be the best. However, overall, $b_c$-BDPC is generally the best, even if some improvements in accuracy may not be significant. There are two reasons for this. One is that most compared methods reported in the literature have chosen or tuned their parameters based on empirical performance. As a result, there would be little room to improve these methods significantly. Another reason is that our proposed methods, $b_c$-BDPC or $k$-BDPC, determine their parameters, $b_c$ and $k$, automatically, without the need for manual tuning. But, the major advantage is that there is no need for tuning these parameters; thus, no robustness issues need to be addressed.

- It should be noted that our developed $b_c$-BDPC and $k$-BDPC are indeed steady algorithms that do not require parameters to be tuned. More specifically, the parameter $b_c$ used in $b_c$-BDPC and the parameter $k$ used in k-BDPC are both determined automatically in Section 2.4.1 without the need for empirical tuning. In this case, there are no robustness issues.

- While BDPC-BS methods have shown promise in BS, there are also some weaknesses. One is that the parameters, $b_c$ for $b_c$-BDPC and $k$ for $k$-BDPC, determined by the proposed automatic rules may not be optimal, as shown in the conducted experiments. Another weakness is that, according to experimental results, $k$-BDPC generally does not perform as well as $b_c$-BDPC. This may be closely related to the value of $k$. In this case, an alternative automatic rule needs be developed for $k$-BDPC.

## 4. Conclusions

This paper presents two BDPC-based BS methods, $b_c$-BDPC and $k$-BDPC, both of which extend current existing DPC-based BS methods. The innovation and originality of this paper can be derived from the following novelties:

(a)  The first and foremost novelty is the introduction of the new concept of band prominence value (BPV) as a third indicator for DPC, which has never been explored in the literature. It can be combined with BLD and BD to yield a new BDPC score, calculated by $\eta = \text{BLD} \times \text{BD} \times \text{BPV}$, which can better rank bands for BS. Experimental results also demonstrate that BPV indeed enhances and further improves BS performance compared to methods that only use BLD and BD for band prioritization.

(b)  A BDPC-BS method, $b_c$-BDPC, is developed. Specifically, an automatic rule is derived for $b_c$-BDPC so that the cut-off band distance $b_c$ can be determined without prior knowledge and empirical determination. In particular, $b_c$ is determined by the number of clusters corresponding to the $n_{\text{classes}}$ used for HSIC.

(c) Analogous with $b_c$-BDPC, another BDPC-BS method, $k$-BDPC, is developed, where an automatic rule is also derived for $k$-BDPC so that the number of nearest neighboring bands, $k$, can be determined by the number of selected bands, $n_{BS}$, which is in turn determined by VD.

(d) In order for BDPC-BS to better capture spectral characteristics, the commonly used Euclidean distance for calculating distance between data points is replaced by three types of spectral discrimination measures, namely SAM, SID, and SIDAM, to calculate spectral correlation among bands. This task cannot be accomplished by DPC-BS methods.

As a final remark, it is interesting to note that the three performance indicators, $\rho_i$ in (1) and $\delta_i$ in (3) used by DPC, coincidentally correspond to Fisher's ratio, within-class distance, and inter-class distance. According to recent work [57], $\rho_i$ in (1), $\delta_i$, and $\gamma_i$ in (5) can be reinterpreted as three class features: intra-class feature (Intra-CF), which measures class variability within a class; inter-class feature (Inter-CF), which measures class separability between classes; and $\gamma_i$ in (5) total class feature (TCF), which multiples Intra-CF and Inter-CF, respectively, for hyperspectral image classification (HSIC). As a consequence, a new concept of Fisher's ratio-based DPC can be derived for HSIC. Efforts along this direction are currently underway.

**Author Contributions:** Conceptualization, C.-I.C.; methodology, C.-I.C. and Y.-M.K.; software, Y.-M.K. and K.Y.M.; validation, Y.-M.K. and K.Y.M.; formal analysis C.-I.C. and Y.-M.K.; investigation, C.-I.C. and Y.-M.K.; writing—C.-I.C.; writing—review and editing, C.-I.C. All authors have read and agreed to the published version of the manuscript.

**Funding:** The work of Chein-I Chang is supported by YuShan Fellow Program, sponsored by the Ministry of Education in Taiwan and also partly supported by the National Science and Technology Council (NSTC) under Grant 111-2634-F-006-012.

**Data Availability Statement:** Data available in a publicly accessible repository that does not issue DOIs. Publicly available datasets were analyzed in this study.

**Conflicts of Interest:** The authors declare no conflicts of interest.

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
