# Peer review of "Band Selection via Band Density Prominence Clustering for Hyperspectral Image Classification"

_remotesensing, doi:10.3390/rs16060942_

Round 1

Reviewer 1 Report

Comments and Suggestions for Authors

This paper introduces BDPC, a novel method for hyperspectral band selection. It employs band local density (BLD), band distance (BD), and band prominence value (BPV) as indicators, expanding DPC-BS. BDPC's two versions, bc-BDPC and k-BDPC, automatically adjust parameters and use spectral discrimination measures, outperforming DPC-BS and other existing BS techniques in various experiments. Several concerns should be addressed before possible publication.

1. While the paper highlights the introduction of novel concepts like Band Prominence Value (BPV) and the development of bc-BDPC and k-BDPC methods, it lacks a robust comparative analysis against existing methods. It would be beneficial to include comparative experiments with other state-of-the-art BS techniques to ascertain the superiority or distinct advantages of the proposed methods.

2. The introduction is lengthy and overly detailed, making it challenging for readers to grasp the main points efficiently. There's a need for a concise and focused overview to introduce the core concepts and motivations without delving deeply into specific methodologies or past research.

3. In addition, the introduction lacks a clear roadmap or structured overview outlining the subsequent sections' content. Breaking down the content into subsections or bullet points could aid readers in understanding the organization and flow of information.
4.
The isolated paragraph in Section 2 can be placed in Section 2.1.

5. The writing format should be uniform, for example, "Fig. 1 ", "Figure 2".

6. Table IV shows the training samples of each class for three data sets. However, the authors didn't gave the dividing criteria of these partitions.

7. Figures 5, 6, 8, 9, 11 and 12 are blurry, especially when enlarged, it is difficult for us to see the details inside.

8. Discussion part: Improving interpretability by quantifying performance differences and explicitly stating the specific performance metrics achieved would enhance the clarity and comprehensibility of these experimental observations. Additionally, further investigations into the underlying reasons for the observed performance disparities could provide deeper insights into the methodological strengths and weaknesses across diverse datasets.

Author Response

This paper introduces BDPC, a novel method for hyperspectral band selection. It employs band local density (BLD), band distance (BD), and band prominence value (BPV) as indicators, expanding DPC-BS. BDPC's two versions, bc-BDPC and k-BDPC, automatically adjust parameters and use spectral discrimination measures, outperforming DPC-BS and other existing BS techniques in various experiments. Several concerns should be addressed before possible publication.

  1. While the paper highlights the introduction of novel concepts like Band Prominence Value (BPV) and the development of bc-BDPC and k-BDPC methods, it lacks a robust comparative analysis against existing methods. It would be beneficial to include comparative experiments with other state-of-the-art BS techniques to ascertain the superiority or distinct advantages of the proposed methods.

Response:

Many thanks for the reviewer’s comments. As much as we would like to do what the reviewer suggested on conducting a robust comparative analysis against existing methods, the matter of fact is that our developed bc-BDPC and k-BDPC are steady algorithms which do not involve parameters needed to be tuned. So, there is no robustness issue. More specifically, the parameter of bc used in bc-BDPC and the parameter of k used in k-BDPC are all determined automatically without tuning empirically. Accordingly, a robust comparative analysis will not be necessary. To address the reviewer’s comments, we have included our above discussions in Section 3.2.4.

  1. The introduction is lengthy and overly detailed, making it challenging for readers to grasp the main points efficiently. There's a need for a concise and focused overview to introduce the core concepts and motivations without delving deeply into specific methodologies or past research.

Response:

Many thanks for the reviewer’s comments. At request, we have streamlined our introduction and moved detailed discussions on methodologies to individual subsections in Section 2 where each of methodologies along with their related works is described in detail. These changes can be seen in the track change version of the revised manuscript.

  1. In addition, the introduction lacks a clear roadmap or structured overview outlining the subsequent sections' content. Breaking down the content into subsections or bullet points could aid readers in understanding the organization and flow of information.

Response:

Many thanks for the reviewer’s comments. At request, we have revised our introduction and broken down 6 subsections to make it concise and easy to follow.

  1. The isolated paragraph in Section 2 can be placed in Section 2.1.

Response:

Many thanks for the reviewer’s comment. We have also reorganized Section 2 as can be seen in the track change version of our revised manuscript.

  1. The writing format should be uniform, for example, "Fig. 1 ", "Figure 2".

Response:

Many thanks for the reviewer’s comment. We have replaced all “Fig.” with “Figure”.

  1. Table IV shows the training samples of each class for three data sets. However, the authors didn't gave the dividing criteria of these partitions.

Response:

Many thanks for the reviewer’s comment. As a matter of fact, the selection of training samples for each class follows exactly the same as that in ref. [52]. To avoid duplicating detailed discussions of selection of training samples, we simply refer it to ref. [52].

  1. Figures 5, 6, 8, 9, 11 and 12 are blurry, especially when enlarged, it is difficult for us to see the details inside.

Response:

Many thanks for the reviewer’s comments. We would like to point out that the figures are generated from Matlab at its best resolution. Nevertheless, if readers want to see more clear plots, they should not have difficulty with seeing details inside the plots since can simply zoom in the figure to find details.

  1. Discussion part: Improving interpretability by quantifying performance differences and explicitly stating the specific performance metrics achieved would enhance the clarity and comprehensibility of these experimental observations. Additionally, further investigations into the underlying reasons for the observed performance disparities could provide deeper insights into the methodological strengths and weaknesses across diverse datasets.

Response:

Many thanks for reviewer’s comments. Following the reviewer’s suggestions, we have revised discussion section, Section 3.2.4 by providing deeper insights into the methodological strengths and weaknesses across diverse datasets section as highlighted by yellow.

Reviewer 2 Report

Comments and Suggestions for Authors

The paper is well written and organised. I have one questions:

Are there other metrics besides the classification results that can be used as evaluation metrics?

Comments on the Quality of English Language

It is ok for representation.

Author Response

The paper is well written and organised. I have one questions:

Are there other metrics besides the classification results that can be used as evaluation metrics?

Response:

Yes, in addition to OA, AA and Kappa coefficient, there is another type of evaluation metric, called precision rate (PR), referred to as a posteriori measure or user’s accuracy, which is based on classified data samples and addresses classification with background [55-56]. Since PR is generally used to evaluate object detection coupled with recall rate in computer vision, its usage and discussions are beyond the scope of this paper. So, no discussions of PR are included. To accommodate the reviewer’s comments, we have included the above discussion in Section 3.2.4 highlighted by yellow.

Comments on the Quality of English Language

It is ok for representation.

Reviewer 3 Report

Comments and Suggestions for Authors

1)     What means (3)? Please add explanations to the text.

2)     Please check (2). dc-dij<0 or >0?

3)     In page 6, line 217, what means “lth band image with L spectral bands”? please check and correct if it is necessary.

4)     BPV calculation described in Fig. 1 and the Algorithm in page 11 is not clear. What does it mean? How does it compute in practice. What is “curve” represented in the “Algorithm for calculation prominence scores of bands”?

5)     There are grammatical errors such as “to determining”.

6)     Please check representation of (45) and (46).

7)     Recent publications (2022-2024) should be cited and discussed.

Comments on the Quality of English Language

There are grammatical errors such as “to determining”.

Author Response

Comments and Suggestions for Authors

  • What means (3)? Please add explanations to the text.

Response:

As requested by the reviewer, we have added the explanation of (3) in the text.

  • Please check (2). dc-dij<0 or >0?

Response:

Many thanks for reviewer’s pointing out the error. It should be dc-dij>0.

  • In page 6, line 217, what means “lth band image with L spectral bands”? please check and correct if it is necessary.

Response:

Many thanks for reviewer’s comment. We have revised the sentence.

  • BPV calculation described in Fig. 1 and the Algorithm in page 11 is not clear. What does it mean? How does it compute in practice. What is “curve” represented in the “Algorithm for calculation prominence scores of bands”?

Response:

Many thanks for reviewer’s comments. We have revised the descriptions of how the curve is calculated.  As a matter of fact, how to calculate prominence of a peak is available in Matlab and can be done by findpeaks on https://www.mathworks.com/help/signal/ug/prominence.html.

  • There are grammatical errors such as “to determining”.

Response:

We have changed “to determining” to “to determine”

  • Please check representation of (45) and (46).

Response:

Both eqs, (45) and (46) are correct.

7)     Recent publications (2022-2024) should be cited and discussed.

Response:

At request, we have included several references from 2022-2024 highlighted by yellow in the reference list.

Comments on the Quality of English Language

There are grammatical errors such as “to determining”.

Response:

We have changed “to determining” to “to determine”

Reviewer 4 Report

Comments and Suggestions for Authors

Introduction

The literature listed in paragraphs 2-7 should be highly summarized and generalized rather than simply listed.

Lines 136-139: References to results should not be written in advance in the introduction.

Conclusion

Please add the shortcomings and prospects of this study.

Table IV: Three-wire format.

Figure 5: The starting scale of the horizontal axis is not displayed.

Try to keep each table in the manuscript to one page; otherwise, repeat the header line.

Line 657: OA = 97.48%. Check the entire manuscript.

Comments on the Quality of English Language

Minor editing of English language required.

Author Response

Comments and Suggestions for Authors

Introduction

The literature listed in paragraphs 2-7 should be highly summarized and generalized rather than simply listed.

Response:

Many thanks for reviewer’s comments. At request, we have revised these paragraphs and moved more detailed descriptions to individual subsections in Section 2. In particular, for each cited reference, we have briefly reviewed its idea.

Lines 136-139: References to results should not be written in advance in the introduction.

Response:

Many thanks for reviewer’s comments. We have removed the references.

Conclusion

Please add the shortcomings and prospects of this study.

Response:

Many thanks for the reviewer’s comment. We have added shortcomings in the discussion section, Section 3.2.4 and prospects of study in conclusion section.

Table IV: Three-wire format.

Response:

Sorry. We are confused with Reviewer’s comment on “three-wire format”. Nevertheless, we have simplified Table IV to make it more clear.

Figure 5: The starting scale of the horizontal axis is not displayed.

Response:

Sorry. We are confused with Reviewer’s comment. According to the figure, the horizontal axis is the band number starting from 0 to 220 as shown in the figure. What is “the starting scale”?.

Try to keep each table in the manuscript to one page; otherwise, repeat the header line.

Response:

At request, we have tried to keep each table in one page.

Line 657: “OA = 97.48%”. Check the entire manuscript.

Response:

Many thanks for reviewer’s comments. We have checked and corrected the erroneous captions in all the figures.

Comments on the Quality of English Language

Minor editing of English language required.

Response:

We have tried our best to make sure that the English paper presentation meets quality standard.

Reviewer 5 Report

Comments and Suggestions for Authors

This paper proposed a new approach for band selection in hyperspectral image based on DPC theory. The integrated BD approach was proved efficient compared with other state-of-the-art algorithms. The clustering theory seems to be beneficial for tackling the issues encountered by other commonly employed techniques.  I have not found any major items to correct in this manuscript. Here (just for indication) some ways to improve the current version of the paper:

1. The main contribution is exploring two versions of BDPC and proposing a new indicator with BLD and BD. The whole process was more like a strategy of several general processes, although it can achieve better results than recently methods. I think the innovation needed to be strengthened and detailed.

2. In the 3 experiments, the improvements of OA and Kappa for the proposed algorithm (bc-BDPC or k-BDPC) are limited. There is not very persuadable. Please give the reason.

3. The manuscript mentioned that the classifiers can not be used as deep learning model in the experiment. However, nowadays, the classifiers based on DL can achieve very high accuracy, which means they are very popular and have a strong applicability in the future. So what is the point of the proposed BD algorithm?

Author Response

This paper proposed a new approach for band selection in hyperspectral image based on DPC theory. The integrated BD approach was proved efficient compared with other state-of-the-art algorithms. The clustering theory seems to be beneficial for tackling the issues encountered by other commonly employed techniques.  I have not found any major items to correct in this manuscript. Here (just for indication) some ways to improve the current version of the paper:

  1. The main contribution is exploring two versions of BDPC and proposing a new indicator with BLD and BD. The whole process was more like a strategy of several general processes, although it can achieve better results than recently methods. I think the innovation needed to be strengthened and detailed.

Response:

Many thanks for the reviewer’s comments. We have included more discussions on innovation as novelties in the conclusion section.

  1. In the 3 experiments, the improvements of OA and Kappa for the proposed algorithm (bc-BDPC or k-BDPC) are limited. There is not very persuadable. Please give the reason.

Response:

Many thanks for the reviewer’s comments. We believe that as shown by experiments our proposed k-BDPC may not be the best but overall speaking, bc-BDPC is generally the best even some improvements on accuracy might not be significant. There are two reasons. One is that most compared methods reported in the literature have chosen or tuned their parameters at the best performance empirically. As a result, there would be little room to improve these methods significantly. Another reason is that our proposed methods, bc-BDPC or k-BDPC determine their parameters, bc and k automatically, in which case, these parameters may not be optimally determined. But the major advantage is that there is no need of tuning these parameters and thus, no robustness issue needs to be addressed. To address the reviewer’s comments, we have included our above discussions in the discussion section, Section 3.2.4.

  1. The manuscript mentioned that the classifiers can not be used as deep learning model in the experiment. However, nowadays, the classifiers based on DL can achieve very high accuracy, which means they are very popular and have a strong applicability in the future. So what is the point of the proposed BD algorithm?

Response:

Many thanks for the reviewer’s comments. We believed that the reviewer may have misunderstood our statement “It turns out that spectral-spatial classifiers meet these requirements and are preferred to deep learning-based classifiers“. What we really meant is that when we selected compared methods, deep learning–based methods were considered. This is because our proposed methods are steady with the parameters bc and k being determined automatically. In this case, spectral-spatial classifiers are more appropriate to be used as classifiers than deep learning-based classifiers. which require a number of model parameters being tuned empirically. To accommodate the reviewer’s comments, we have included the above discussions in the text. As mentioned above, our proposed bc-BDPC and k-BDPC are automatic with on parameters needed to be tuned. This is a significant advantage compared to DL-based methods.  

Round 2

Reviewer 1 Report

Comments and Suggestions for Authors

Many thanks to the authors for revising their manuscript. The paper is ready for publication.